# Neuronal junctophilins recruit specific Ca$_V$ and RyR isoforms to ER-PM junctions and functionally alter Ca$_V$2.1 and Ca$_V$2.2

**Stefano Perni, Kurt Beam\***

Department of Physiology and Biophysics, Anschutz Medical Campus, University of Colorado, Aurora, United States

**Abstract** Junctions between the endoplasmic reticulum and plasma membrane that are induced by the neuronal junctophilins are of demonstrated importance, but their molecular architecture is still poorly understood and challenging to address in neurons. This is due to the small size of the junctions and the multiple isoforms of candidate junctional proteins in different brain areas. Using colocalization of tagged proteins expressed in tsA201 cells, and electrophysiology, we compared the interactions of JPH3 and JPH4 with different calcium channels. We found that JPH3 and JPH4 caused junctional accumulation of all the tested high-voltage-activated Ca$_V$ isoforms, but not a low-voltage-activated Ca$_V$. Also, JPH3 and JPH4 noticeably modify Ca$_V$2.1 and Ca$_V$2.2 inactivation rate. RyR3 moderately colocalized at junctions with JPH4, whereas RyR1 and RyR2 did not. By contrast, RyR1 and RyR3 strongly colocalized with JPH3, and RyR2 moderately. Likely contributing to this difference, JPH3 binds to cytoplasmic domain constructs of RyR1 and RyR3, but not of RyR2.

**\*For correspondence:**
kurt.beam@cuanschutz.edu

## Introduction

Close appositions between the endoplasmic reticulum and plasma membrane ('ER-PM junctions') play important roles in membrane trafficking and cellular signaling within diverse tissues (*Porter and Palade, 1957*; *Rosenbluth, 1962*; *Gardiner and Grey, 1983*; *Poburko et al., 2004*; *Wu et al., 2006*), including neurons where there are large numbers of such junctions (*Wu et al., 2017*). Among the proteins that can induce the formation of ER-PM junctions are the junctophilins ('JPH'), which have a single, short C-terminal segment that traverses the ER membrane and which bind to the inner surface of the plasma membrane by virtue of repeated 'MORN motifs that are located toward the N-terminus (*Takeshima et al., 2000*).

Arguably, the most extensively characterized ER-PM junctions are those in skeletal and cardiac muscle. In both cell types, these junctions (triads and dyads) are the site at which functional coupling occurs between high-voltage-activated calcium channels in the plasma membrane (skeletal: Ca$_V$1.1; cardiac: Ca$_V$1.2) and calcium release channels in the ER (skeletal: RyR1; cardiac: RyR2). The formation of these junctions depends upon JPH2 in cardiac muscle and on both JPH2 and JPH1 in skeletal muscle (*Takeshima et al., 2000*; *Ito et al., 2001*). The closely related proteins, JPH3 and JPH4, are expressed in the nervous system (*Nishi et al., 2003*). Knockout of JPH4 does not produce an overt behavioral phenotype (*Moriguchi et al., 2006*), whereas knockout of only JPH3 causes motor discoordination (*Nishi et al., 2002*) that seemed to progress with aging (*Seixas et al., 2012*). However, a broad spectrum of changes occurs when both JPH3 and JPH4 are knocked out, including an aberrant hindlimb reflex, altered salivary secretion, and impaired memory (*Moriguchi et al., 2006*). Moreover, a triplet repeat expansion in exon 2A of human JPH3 causes reduced expression of JPH3 and results in Huntington disease-like 2 (*Seixas et al., 2012*). Thus, junctophilin-containing ER-PM junctions appear to be important for multiple neuronal functions.

Transcripts of JPH3 and JPH4 are present in diverse brain regions, with expression being highest in the hippocampal formation, isocortex, cortical subplate, striatum, and olfactory systems (*Nishi et al., 2003*; *Lein et al., 2007*). Groups of cells in other regions (e.g., cerebellar granule cells) also show high expression (*Nishi et al., 2003*; *Lein et al., 2007*). However, the electrophysiological consequences of the double knockout of JPH3 and JPH4 have only been investigated in slice recordings of hippocampal CA1 neurons and cerebellar Purkinje cells. In CA1 pyramidal neurons, a single action potential is followed by an afterhyperpolarization (AHP), which was found to be greatly reduced by apamin, SERCA inhibition, and ryanodine, and to be absent in neurons from JPH3/JPH4 double KO mice (*Moriguchi et al., 2006*). A similar pharmacological profile and absence in double KO cells was found for the slow AHP following complex spikes in Purkinje neurons (*Kakizawa et al., 2007*), which is perhaps surprising given that the expression of JPH3 and JPH4 is relatively low in Purkinje cells (*Nishi et al., 2003*). Based on their observations, *Moriguchi et al., 2006* and *Kakizawa et al., 2007* proposed that the AHP in CA1 neurons, and slow AHP in Purkinje cells, were generated at ER-PM junctions, which contained the neuronal junctophilins together with voltage- or ligand-activated $Ca^{2+}$ channels and SK $Ca^{2+}$-activated potassium channels in the plasma membrane and ryanodine receptors in the ER.

The molecular components of ER-PM junctions in CA1 neurons have also been probed with biochemical and immunolabeling approaches (*Kim et al., 2007*; *Sahu et al., 2019*). In particular, *Kim et al., 2007* found that $Ca_V1.3$ and RyR2 interacted with one another and colocalized in CA1 neurons. Subsequently, *Sahu et al., 2019* found with ultra-resolution microscopy that these neurons contained clusters of ryanodine receptors (specifically RyR2), SK channels (specifically KCa3.1), and L-type $Ca^{2+}$ channels ($Ca_V1.2$ and $Ca_V1.3$), and found that this channel complex clusters with JPH3. Thus, the case can be made that the AHP in CA1 neurons can be generated at JPH3-containing ER-PM junctions by (1) $Ca^{2+}$ entry across the plasma membrane via L-type channels, (2) $Ca^{2+}$ release via RyR2 in the ER, and (3) activation of SK channels in the plasma membrane (*Sahu et al., 2019*), with entry of $Ca^{2+}$ via NMDA receptors also being of importance (*Moriguchi et al., 2006*).

Although a picture is emerging about junctophilins in CA1 neurons, many questions remain unanswered about the neuronal junctophilins in other neuronal populations. For example, the presence of RyR2 in ER-PM junctions of hippocampal CA1 neurons is consistent with its being the most prominent RyR isoform in those cells, based on in situ hybridization (*Mori et al., 2000*). However, the RyR1 signal is comparable to, or exceeds that of, RyR2 in the hippocampal dentate gyrus, mitral cells of the olfactory bulb, olfactory tubercle, and cerebellar Purkinje cells (*Mori et al., 2000*). Moreover, RyR3 is a significant fraction of, or exceeds, RyR2 in hippocampal CA1 neurons and olfactory tubercle (*Mori et al., 2000*). Thus, the question arises: is relative expression the only factor influencing the junctional recruitment of RyR isoforms by the neuronal junctophilins or is there preferential recruitment of one or more isoforms?

There are similar questions about which voltage-gated calcium channels are recruited by the neuronal junctophilins. For example, both P/Q ($Ca_V2.1$) and T-type ($Ca_V3.1$) channels are highly expressed in cerebellar Purkinje cells (*Lein et al., 2007*). However, only $Ca^{2+}$ influx through the P/Q type causes activation of the SK channels that produces the AHP following simple spikes (*Womack et al., 2004*). An obvious question based on these results, and those of *Kakizawa et al., 2007*, is whether the neuronal junctophilins cause junctional recruitment of $Ca_V2.1$ and not of $Ca_V3.1$. A related question is whether $Ca_V2.2$ is recruited by JPH3 and/or JPH4, a possibility that has not been suggested by previous studies of the junctophilins. Nonetheless, the recruitment of $Ca_V2.2$ seems plausible because $Ca_V2.2$ (*Lein et al., 2007*), JPH3, and JPH4 (*Nishi et al., 2003*) are all expressed at moderate levels in the paraventricular nucleus of the thalamus.

The goal of the work described here was to address the ability of the neuronal junctophilins to cause the junctional accumulation of specific $Ca_V$ and RyR isoforms. An additional goal was to determine whether such junctional accumulation differs between JPH3 and JPH4 since such a difference could help explain why single knockout of JPH3, but not JPH4, results in a neurological phenotype (*Moriguchi et al., 2006*). For this work, we chose to use heterologous expression in tsA201 cells to provide a uniform basis for comparing the $Ca_V$, RyR, and JPH isoforms. Our work demonstrates that JPH3 and JPH4 both fail to cause $Ca_V3.1$ to accumulate at junctions, but that both cause the accumulation of P/Q ($Ca_V2.1$) and N ($Ca_V2.2$) channels. In addition, both junctophilins cause a slowing of the inactivation of $Ca_V2.1$ and $Ca_V2.2$. JPH3 and JPH4 differ substantially in their ability to recruit RyRs. Whereas JPH4 causes some accumulation of RyR3, JPH3 causes accumulation of all three RyR

isoforms, especially RyR1. An important contributor to the accumulation of RyR1 arises from a binding interaction between its cytoplasmic domain and a region of JPH3 that is highly divergent from the corresponding region of JPH4.

## Results

### Fluorescently tagged JPH3 and JPH4 induce the formation of ER-PM junctions in tsA201 cells

In order to compare the two neuronal junctophilins (*Figure 1A*) in their ability to recruit voltage-gated calcium channels and ryanodine receptors, we began by verifying that attachment of a fluorescent protein at the amino terminus did not interfere with their ability to induce ER-PM junctions in tsA201 cells. As shown in *Figure 1B*, confocal sections through the middle of the cell (top row) revealed that both mCherry-JPH3 and mCherry-JPH4 were arrayed as discrete fluorescent segments outlining the cell periphery, and confocal sections at the substrate-adhering surface of the cell (middle row) revealed small fluorescent patches (diameter of ~ 0.5 to 2 μm), which were generally larger for JPH4. Thin section electron microscopy confirmed the presence of ER-PM junctions in cells transfected with mCherry-JPH3 or mCherry-JPH4 (*Figure 1B*, bottom row). We did not carry out a quantitative analysis of the thin sections obtained from cells transfected with JPH3 or JPH4. However, our previous analysis of thin sections demonstrated that ER-PM junctions in non-transfected tsA201 cells are infrequent ($\geq$ 2 junctions in 8 % of cells, with a maximum of three junctions) and of small size (mean length of ~ 0.15 μm, maximum of 0.28 μm) (*Perni et al., 2017*). By way of comparison, two or more junctions were present in ~ 20 % of the cells transfected with a neuronal junctophilin (about the same as the transfection efficiency). These junctions were also substantially longer than those in non-transfected cells (the ones shown in *Figure 1B*, bottom panels, are roughly 0.55 and 0.77 μm for JPH3 and JPH4, respectively). Additionally, the junctions induced by JPH3 or JPH4 had a relatively uniform gap of about ~ 7 nm separating the ER and PM, which is similar to the gap in ER-PM junctions of dyspedic (RyR1-null) myotubes (*Takekura et al., 1995*), which endogenously express JPH2 (*Felder et al., 2002*).

### JPH3 and JPH4 recruit Ca$_V$2.1 and Ca$_V$2.2 to ER-PM junctions

Although gene knockout and electrophysiology provide evidence that Ca$_V$2.1 channels are present in ER-PM junctions induced by the neuronal junctophilins (*Kakizawa et al., 2007*; *Womack et al., 2004*), such localization has not been tested using the fluorescent microscopy techniques that have been applied to the L-type channels Ca$_V$1.2 and Ca$_V$1.3 (*Sahu et al., 2019*). Moreover, the possibility that Ca$_V$2.2 might be in such junctions does not appear to have been previously considered. Here, we have tested whether Ca$_V$2.1 and Ca$_V$2.2 are recruited to junctions by JPH3 and/or JPH4, comparing them to Ca$_V$1.2, which was previously shown to associate with both these neuronal junctophilins (*Sahu et al., 2019*). We also tested the low-voltage-activated (LVA) Ca$^{2+}$ channel, Ca$_V$3.1. Expressed without junctophilins, all these Ca$^{2+}$ channels (*Figure 2A*, top row) were distributed in a reticular pattern in the cell interior and distributed in variable patterns near the cell surface. Co-expressed with either JPH3 (*Figure 2B*) or JPH4 (*Figure 2C*), the three high-voltage-activated (HVA) channels became concentrated in discrete regions that colocalized at the cell periphery with the junctophilins, whereas Ca$_V$3.1 did not. Note that both here and in subsequent figures the channels are always represented in green and the junctophilins in red so that areas of overlap appear yellow in the merged images.

In order to quantify the extent to which the channels at the cell periphery were colocalized with the junctophilins, it was necessary to minimize the contribution of channels present in reticular structures within the cell interior. For this, we obtained ~ 0.9- μm-thick optical scans of the bottom surface of the cell. These bottom-surface scans were then used to calculate Pearson's colocalization coefficients (*Figure 2D–G*). For the HVA channels, the mean colocalization coefficients ranged from 0.8 (Ca$_V$2.2/JPH4) down to 0.50 (Ca$_V$2.1/JPH3). The means for Ca$_V$3.1 (0.25 vs. JPH3 and 0.35 versus JPH4) were significantly (p < 0.0001) smaller than those for the HVA channels (Ca$_V$1.2, Ca$_V$2.1, and Ca$_V$2.1) versus either JPH3 or JPH4. One implication of these results is that the neuronal junctophilins recruit the P/Q channel Ca$_V$2.1 to ER-PM junctions in preference to the T-type channel Ca$_V$3.1, which is in good agreement with earlier results demonstrating that the AHP in Purkinje cells is

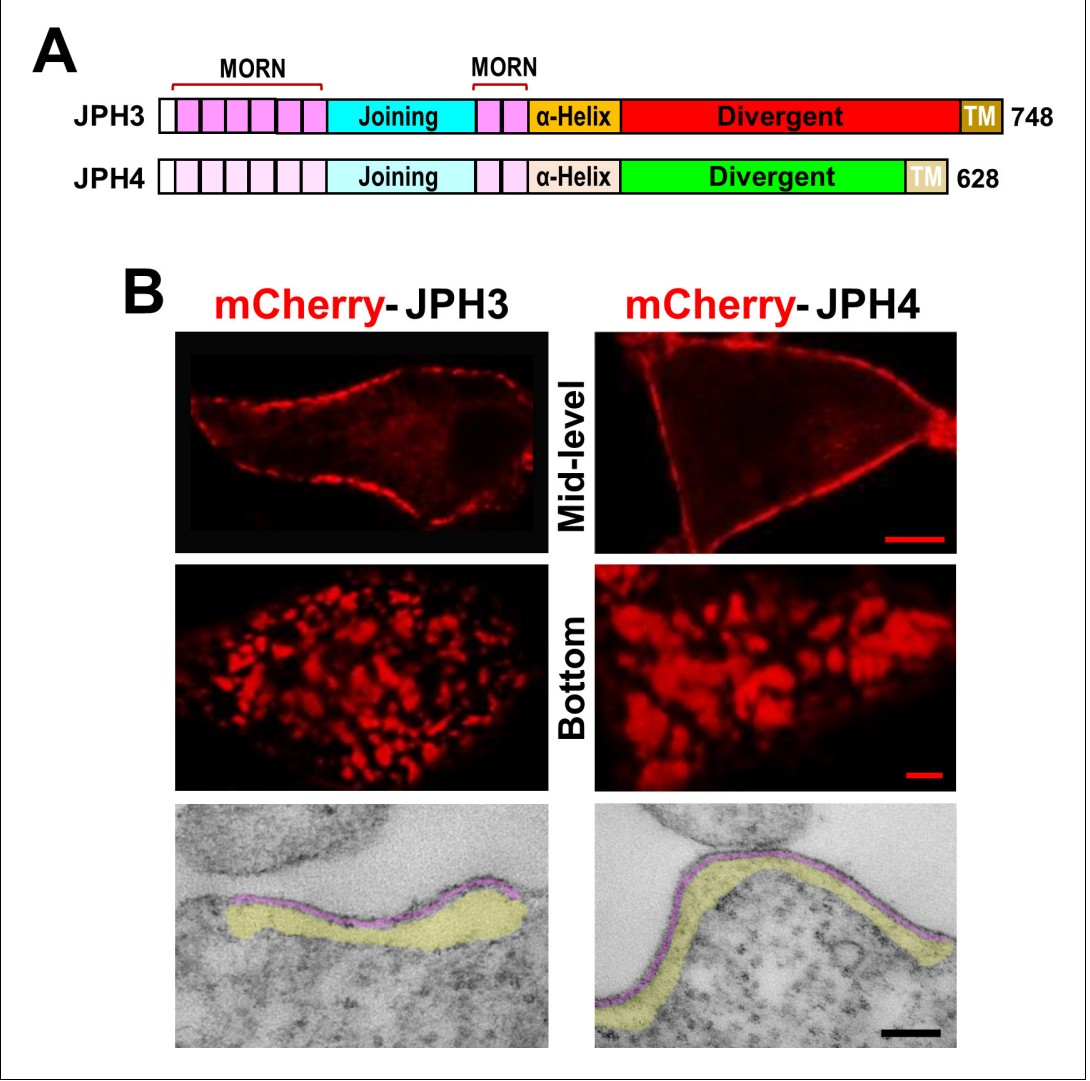

**Figure 1.** Junctions between the endoplasmic reticulum and plasma membrane (ER-PM junctions) are induced in tsA201 cells by expression of JPH3 or JPH4 N-terminally tagged with fluorescent proteins. (**A**) Schematic representation of JPH3 and JPH4, indicating the 'MORN' motifs that bind to the plasma membrane and C-terminal segment ('TM') that traverses the ER membrane. Except for the 'Divergent' domain, the two proteins display substantial sequence similarity (*Garbino et al., 2009*). (**B**) Confocal optical sections acquired at mid-level (top row) or the bottom surface (middle row), and thin section electron micrographs (bottom row), are shown for cells transfected with mCherry-JPH3 or mCherry-JPH4 (left and right columns, respectively). The fluorescence was predominantly present in discrete foci near the cell periphery as expected for ER-PM junctions, which can be directly visualized in the electron micrographs in which the junctional gap and the ER sub-cortical cisternae are pseudo-colored in purple and yellow, respectively. Scale bars = 5 µm (top row), 2 µm (middle row), and 100 nm (bottom row).

preferentially triggered by P/Q channels (*Womack et al., 2004*) and that this AHP is greatly reduced when the two neuronal junctophilins are knocked out (*Kakizawa et al., 2007*).

## JPH3 and JPH4 significantly slow the inactivation of Ca$_V$2.1 and Ca$_V$2.2

To determine whether trafficking to junctions affected current density or channel properties, whole-cell clamping was used to measure calcium currents in cells expressing Ca$_V$1.2, Ca$_V$2.1, and Ca$_V$2.2 with, or without, JPH3 or JPH4. Neither JPH3 nor JPH4 caused large shifts in the peak I–V relationships for any of these three channels (*Figure 3A–C*, top panels), indicating that the voltage dependence of activation was little affected. Macroscopic current densities were essentially unchanged for

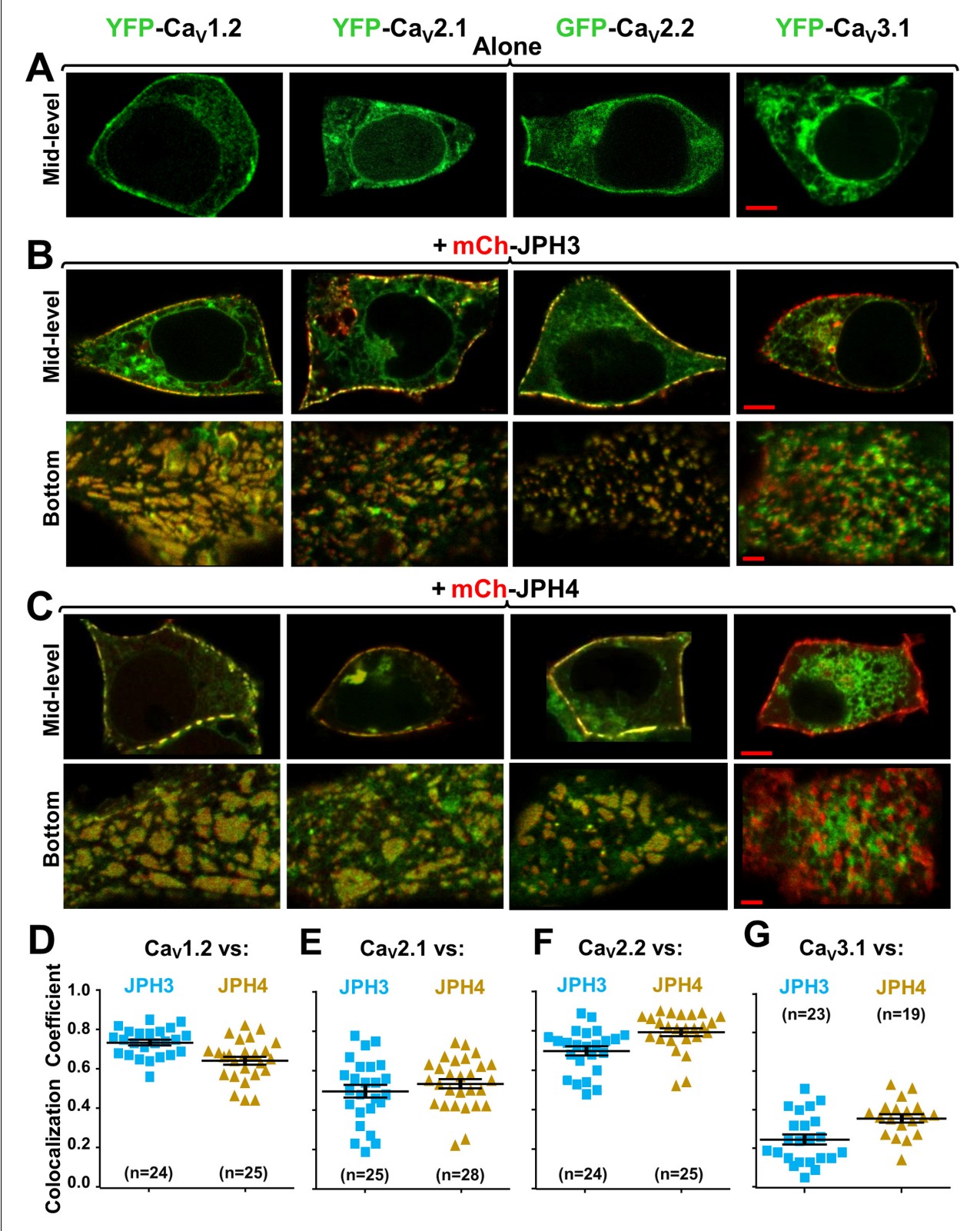

**Figure 2.** Three neuronal, high-voltage-activated calcium channels (Ca$_V$1.2, Ca$_V$2.1, and Ca$_V$2.2), but not a low-voltage-activated one (Ca$_V$3.1), localize at junctions induced between the endoplasmic reticulum and plasma membrane by JPH3 or JPH4. Mid-level or bottom-surface optical sections are shown for tsA201 cells transfected with the designated Ca$_V$ constructs (represented in green) either in the absence of junctophilins (**A**) or together with either mCherry-JPH3 (**B**) or mCherry-JPH4 (**C**), in which the junctophilins are represented in red in merged red/green images. cDNAs for the auxiliary subunits

*Figure 2 continued on next page*

*Figure 2 continued*

β1b and α2-δ1 were also present for the high-voltage-activated channels. For the three high-voltage-activated channels, co-expression with JPH3 or JPH4 resulted in clusters of channels that were near the surface and colocalized with the junctophilins. Conversely, $Ca_V3.1$ localization was unaffected by the presence of the two JPHs. Scale bars = 5 and 2 µm, respectively, for the mid-level and bottom-surface images. (D–G) Pearson's colocalization coefficients for the specified combinations of neuronal calcium channels and junctophilins, which were calculated from optical sections of the bottom of the cell that was adjacent to the substrate (see B and C for examples). In this plot (and subsequent plots), individual data points indicate Pearson's coefficient for a single cell, with the mean and ± SEM for each construct combination indicated by longer and shorter horizontal lines, respectively. Numbers of cells are indicated in parentheses. Pearson's coefficients and their statistical comparison are provided in *Figure 2—source data 1*. The online version of this article includes the following source data for figure 2:

**Source data 1.** Numerical data and statistical analyses to support graphs in *Figure 2*.

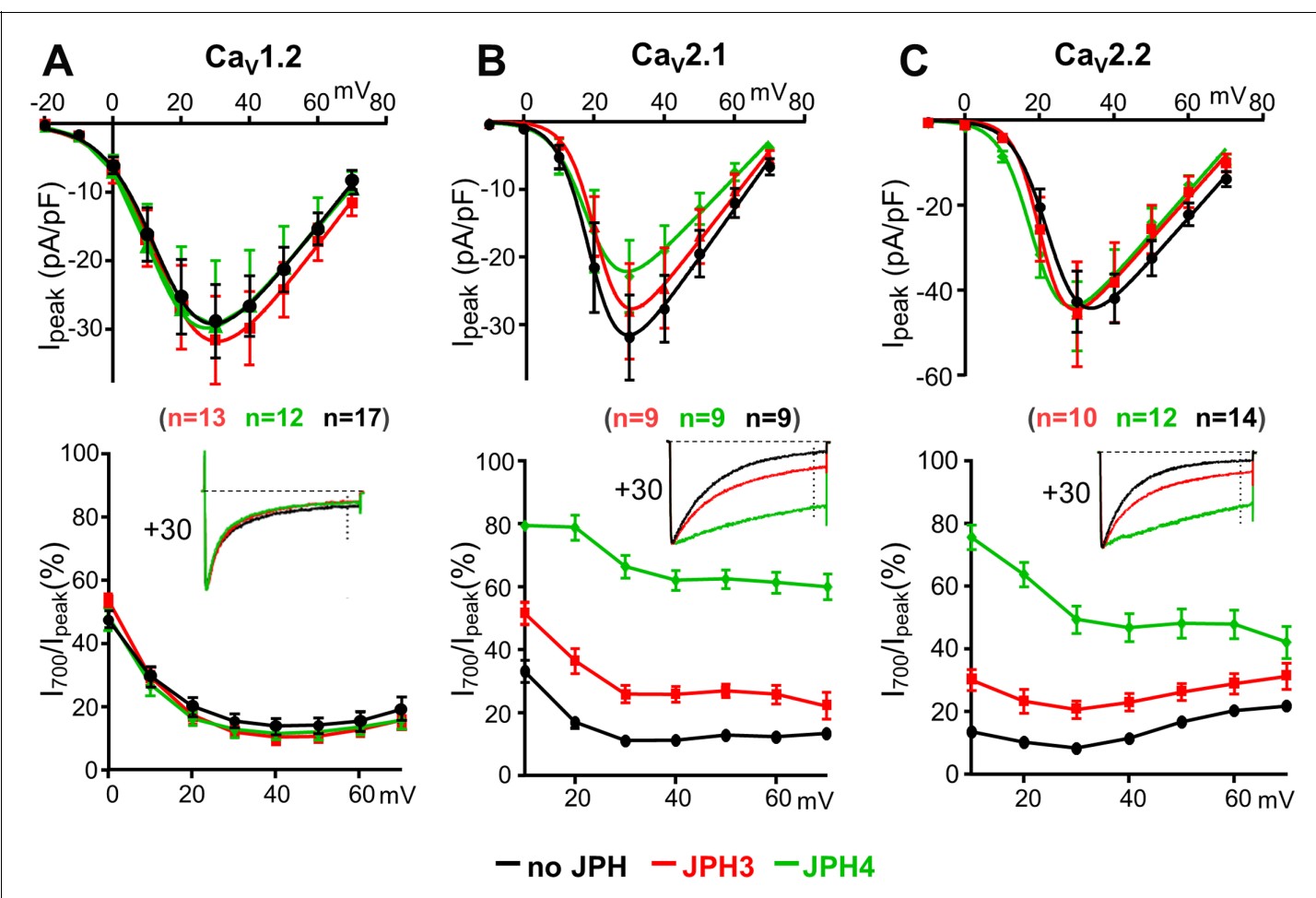

**Figure 3.** JPH3 and JPH4 slow the inactivation of $Ca^{2+}$ currents via $Ca_V2.1$ and $Ca_V2.2$ but not those via $Ca_V1.2$. $Ca^{2+}$ currents were measured in tsA201 cells transfected with $Ca_V1.2$ (A), $Ca_V2.1$ (B), or $Ca_V2.2$ (C) either without junctophilins (black) or together with either JPH3 (red) or JPH4 (green). Constructs for β1b and α2-δ1 were also present. The upper row of panels illustrates the average peak current versus voltage relationships and the lower row of panels plots the percentage of peak current remaining 700 ms after the peak ($I_{700}/I_{peak}$) as a function of test potential. The insets illustrate representative currents (scaled to match in height) elicited by an 800 ms depolarization to the indicated potential, with the current 700 ms after the peak indicated by the vertical dotted line. Data are shown as mean ± SEM. Tables of $I_{peak}$ and $I_{700}$ are provided in *Figure 3—source data 1*. The online version of this article includes the following source data and figure supplement(s) for figure 3:

**Source data 1.** Numerical data to support graphs in *Figure 3*.

**Figure supplement 1.** Similar to its effects on $Ca^{2+}$ currents, JPH4 slows inactivation of $Ba^{2+}$ currents via $Ca_V2.1$ and $Ca_V2.2$, with less effect on inactivation of $Ba^{2+}$ currents via $Ca_V1.2$.

**Figure supplement 1—source data 1.** Numerical data to support graphs in *Figure 3—figure supplement 1*.

$Ca_V1.2$ and $Ca_V2.2$ but were slightly reduced for $Ca_V2.1$. The cause of this reduced current density (decreased membrane expression and/or reduced open probability) was not investigated.

The neuronal junctophilins had little effect on the inactivation of calcium currents via the L-type channel $Ca_V1.2$ (*Figure 3A*, lower panel) but slowed the inactivation of calcium currents via the non-L-type channels, $Ca_V2.1$ and $Ca_V2.2$ (*Figure 3B, C*, lower panels), with JPH4 having a greater effect than JPH3. Given its larger effect, we also tested whether JPH4 affected inactivation of barium currents. As for calcium currents, the inactivation of barium currents via $Ca_V2.1$ and $Ca_V2.2$ was greatly slowed by JPH4, which had less effect on barium currents via $Ca_V1.2$ (*Figure 3—figure supplement 1*).

To ascertain whether the slowing of inactivation required the formation of ER-PM junctions, we also tested the effects of JPH3(1 – 707) and JPH4(1 – 576), which lack the small segment of the C-terminus that traverses the ER membrane (*Figure 1A*) and thus retain the ability to associate with the surface membrane (leftmost panels of Figure 10A, B) but not to induce ER-PM junctions. Nonetheless, JPH3(1 – 707) and JPH4(1 – 576) slowed inactivation of $Ca_V2.1$ and $Ca_V2.2$ (*Figure 4A, B*) to an extent that was similar to, or greater than, that of the full-length constructs (*Figure 3A, B*). JPH4 (1 – 576) slowed the inactivation of $Ca_V2.1$ and $Ca_V2.2$ (*Figure 4A, B*) to an extent that was similar to that of full-length JPH4 (*Figure 3B, C*), whereas JPH3(1 – 707) produced a slowing of inactivation that was greater than that of full-length JPH3, especially for $Ca_V2.1$ (*Figure 3B, C*, *Figure 4A, B*). These results are consistent with the possibility that the slowing of inactivation is a consequence of a direct interaction between the channels and junctophilins, and that truncated JPH3 interacts with a larger fraction of $Ca_V2.1$ channels in the membrane than does full-length JPH3.

## JPH3 is substantially more effective at recruiting RyRs to ER-PM junctions than JPH4

*Figure 5* compares the colocalization of the three RyR isoforms after co-expression in tsA201 cells with either JPH3 (*Figure 5A*) or JPH4 (*Figure 5B*). The corresponding Pearson's coefficients (*Figure 5C*) reveal that for JPH3 the highest colocation occurred with RyR1, with slightly less colocalization with RyR3, and significantly less with RyR2. For JPH4, the greatest colocalization occurred with RyR3, whereas there was significantly less colocalization with both RyR1 and RyR2. The colocalization of RyR3 with JPH4 was not significantly (p = 0.98) different from the colocalization of RyR2 with JPH3. However, for any given RyR isoform, the colocalization with JPH4 was significantly (p < 0.0001) lower than with JPH3. Based on these results, it seems that the greater effectiveness of JPH3 in causing the accumulation of RyRs may help to explain why single knockout of JPH3 affects neurological function, whereas single knockout of JPH4 does not.

Based on previous work (*Sahu et al., 2019*), it seemed possible that RyRs endogenously expressed in tsA201 cells may have influenced the colocalization between the expressed RyRs and junctophilins. To assess the levels of RyR expression, we used Fluo8 to evaluate changes in cytoplasmic calcium in response to the application of the RyR activator, caffeine. The application of caffeine caused large calcium transients in tsA201 cells stably transfected with RyR1 but not in control cells (*Figure 5—figure supplement 1*), in which the fluorescence changes over time were essentially the same as in cells not exposed to caffeine. Thus, it appears that the levels of endogenously expressed RyRs were negligible compared to those of the heterologously expressed RyRs.

### Tripartite junctions
Having examined the ability of the neuronal junctophilins to cause either the accumulation of voltage-gated calcium channels in the plasma membrane or the accumulation of ryanodine receptors in the ER, we next investigated joint recruitment of voltage-gated calcium channels and ryanodine receptors. One question was whether the moderate colocalization of RyR2 with JPH3, and of RyR3 with JPH4 (*Figure 5*), is increased by the additional presence of $Ca_V1.2$. $Ca_V1.2$ was selected because it is recruited to junctions by both JPH3 and JPH4 (*Figure 2*), is known to interact functionally with RyR2 in both neurons (*Sahu et al., 2019*) and cardiomyocytes (*Sham et al., 1995*), and is highly expressed together with RyR3 in extraocular muscle (*Sekulic-Jablanovic et al., 2015*). *Figure 6* compares Pearson's coefficients determined for JPH3 versus RyR2 in the absence and presence of $Ca_V1.2$ and for JPH4 versus RyR3, also in the absence and presence of $Ca_V1.2$. The

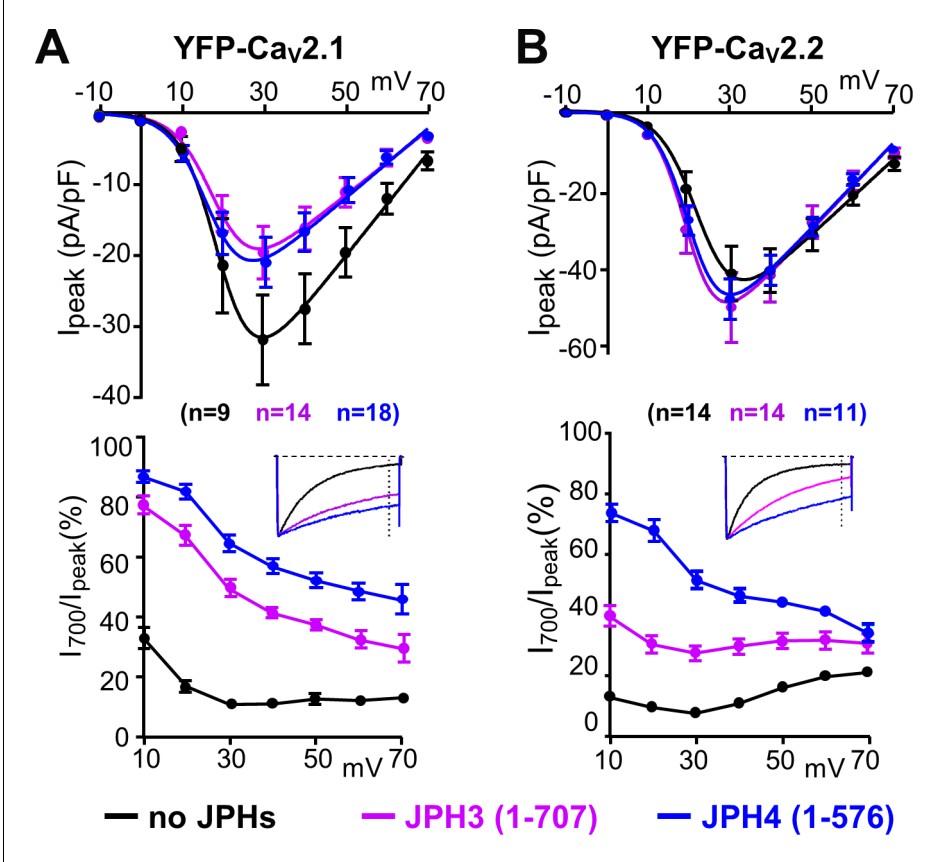

**Figure 4.** The ability of JPH3 and JPH4 to slow inactivation of $Ca_V2.1$ and $Ca_V2.2$ does not depend upon the formation of junctions induced between the endoplasmic reticulum and plasma membrane. Cells were transfected with the calcium channels together with JPH3(1 – 707) or JPH4(1 – 576), which lack the ER-spanning membrane segment and thus associate with the cell surface without inducing ER-PM junctions. (**A, B**) Peak current and $I_{700}/I_{peak}$ as a function of test potential for $Ca_V2.1$ and $Ca_V2.2$, respectively, expressed without junctophilin, with JPH3(1 – 707) or JPH4(1 – 576), indicated in black, purple, and blue, respectively. cDNAs for the $Ca_V$ auxiliary subunits β1b and α2-δ1 were also present. The insets illustrate representative $Ca^{2+}$ currents, which were elicited by 800 ms depolarizations to +30 mV and scaled to match in peak height. On average, the truncated junctophilins slowed inactivation to an extent that was comparable to, or greater than, that caused by the full-length junctophilins (*Figure 3*). Data are shown as mean ± SEM. Tables of $I_{peak}$ and $I_{700}$ are provided in *Figure 4—source data 1*. The online version of this article includes the following source data for figure 4:

**Source data 1.** Numerical data to support graphs in *Figure 4*.

additional presence of $Ca_V1.2$ did not have a significant effect either on the colocalization of RyR2 with JPH3 or on the colocalization of RyR3 with JPH4.

*Figure 7A-D* illustrates tsA201 cells which were co-transfected with mCherry-tagged JPH3 or JPH4, with GFP-tagged $Ca_V2.1$ or $Ca_V2.2$, and with CFP-tagged RyR1. For $Ca_V2.1$, representative cells are displayed in *Figure 7A*, and Pearson's colocalization coefficients are plotted in *Figure 7B*. These indicate that JPH3 caused $Ca_V2.1$ to colocalize with RyR1, consistent with its ability to recruit not only $Ca_V2.1$ (*Figure 2*), but also RyR1 (*Figure 5*). Contrastingly, JPH4, which recruited $Ca_V2.1$ (*Figure 2*) but not RyR1 (*Figure 5*), failed to cause colocalization of $Ca_V2.1$ with RyR1 (*Figure 7A, B*). A similar pattern was observed for $Ca_V2.2$: JPH3, but not JPH4, caused $Ca_V2.2$ to colocalize with RyR1 (*Figure 7C, D*).

To analyze calcium movements in cells expressing JPH3 and RyR1 together with $Ca_V2.1$ or $Ca_V2.2$, we monitored cytoplasmic calcium with Fluo8 and calcium in the ER with R-CEPIAer. Because the green and red fluorescence of these two indicators interfered with using fluorescent protein tags to ensure the simultaneous presence of the necessary proteins, the experiments were

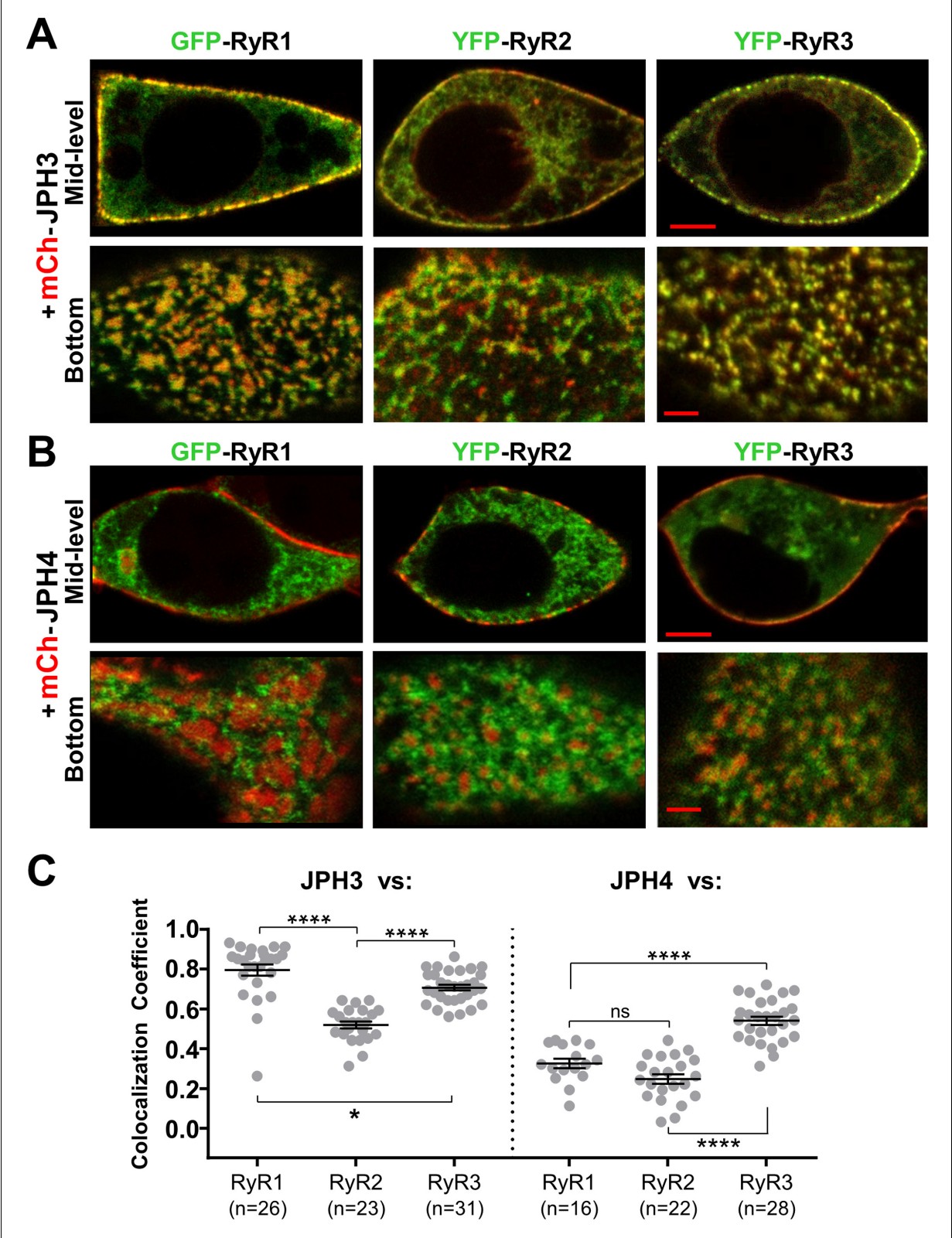

**Figure 5.** JPH3 recruits all three RyR isoforms to junctions between the endoplasmic reticulum and plasma membranes, whereas JPH4 only recruits RyR3. Representative, red/green merged images of the mid-level or bottom surface of tsA201 cells expressing GFP-RyR1, YFP-RyR2, or YFP-RyR3 (left to right, represented in green) together with either mCherry-JPH3 (**A**) or mCherry-JPH4 (**B**), which are represented in red. Scale bars = 5 and 2 μm, respectively, for the mid-level and bottom-surface images. (**C**) Pearson's colocalization coefficients for the specified combinations of junctophilins and

*Figure 5 continued on next page*

*Figure 5 continued*

RyRs, which were calculated from bottom-surface optical sections. Statistical significance: ****p < 0.0001, *p = 0.0248, p = 0.2366 (ns). Pearson's coefficients and their statistical comparison are provided in *Figure 5—source data 1*.

The online version of this article includes the following source data and figure supplement(s) for figure 5:

**Source data 1.** Numerical data and statistical analyses to support graphs in *Figure 5*.
**Figure supplement 1.** Levels of RyRs endogenously expressed in tsA201 cells are very low.
**Figure supplement 1—source data 1.** Numerical data to support graph in *Figure 5—figure supplement 1*.

carried out in HEK293 cells stably transfected with RyR1 and transiently transfected with CFP-JPH3, R-CEPIAer, the voltage-gated calcium channels and their auxiliary subunits. After being loaded with Fluo8-AM, cells were initially selected for experimentation based on the presence of both JPH3 and R-CEPIAer as indicated by cyan and red fluorescence, respectively. The fluorescence signals of Fluo8 and R-CEPIAer were then measured in response to depolarization induced by the application of a solution containing elevated potassium. *Figure 7E, F* illustrates a representative cell in which the transfected calcium channel was Ca$_V$2.1. The yellow outline superimposed on the images of the cell at rest (*Figure 7E*) indicates a region at the cell periphery in which fluorescence changes were measured over time. Depolarization caused a rapid increase in the fluorescence of the cytoplasmic calcium indicator, Fluo8 (*Figure 7F*, green trace), which indicates that Ca$_V$2.1 was present in the plasma membrane. In this cell, there was also a slower decrease in the fluorescence of the ER calcium indicator, R-CEPIAer (*Figure 7F*, red trace). Thus, the behavior of this cell is consistent with the hypothesis that calcium entry via Ca$_V$2.1 triggered the release of ER calcium via RyR1. Altogether, of the 12 similarly transfected cells that produced a Fluo8 transient indicative of the presence of Ca$_V$2.1, 4 also displayed an R-CEPIAer transient demonstrating ER calcium release. The average Fluo8 and R-CEPIAer transients for these four cells are illustrated in *Figure 7G*.

*Figure 7H–J* illustrates results from RyR1-stable cells that were transfected with CFP-JPH3, R-CEPIAer, and Ca$_V$2.2 plus its auxiliary subunits. Out of 11 of these cells producing a Fluo8 transient in response to KCl depolarization, 4 also displayed an R-CEPIAer transient demonstrating ER calcium

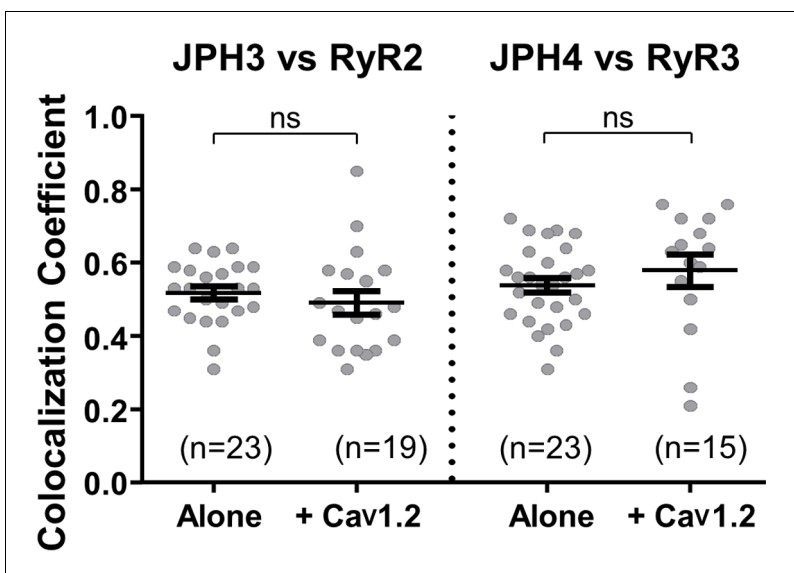

**Figure 6.** The additional expression of Ca$_V$1.2 does not affect colocalization between JPH3 and RyR2 or between JPH4 and RyR3. Pearson's coefficients, calculated from bottom-surface scans, for mCherry-JPH3 versus YFP-RyR2 (left), and for mCherry-JPH4 versus YFP-RyR3 (right) expressed either alone or together with CFP-Ca$_V$1.2 plus β1b and α2-δ1. Statistical significance: p > 0.4 (ns). Pearson's coefficients and their statistical comparison are provided in *Figure 6—source data 1*.

The online version of this article includes the following source data for figure 6:

**Source data 1.** Numerical data and statistical analyses to support graphs in *Figure 6*.

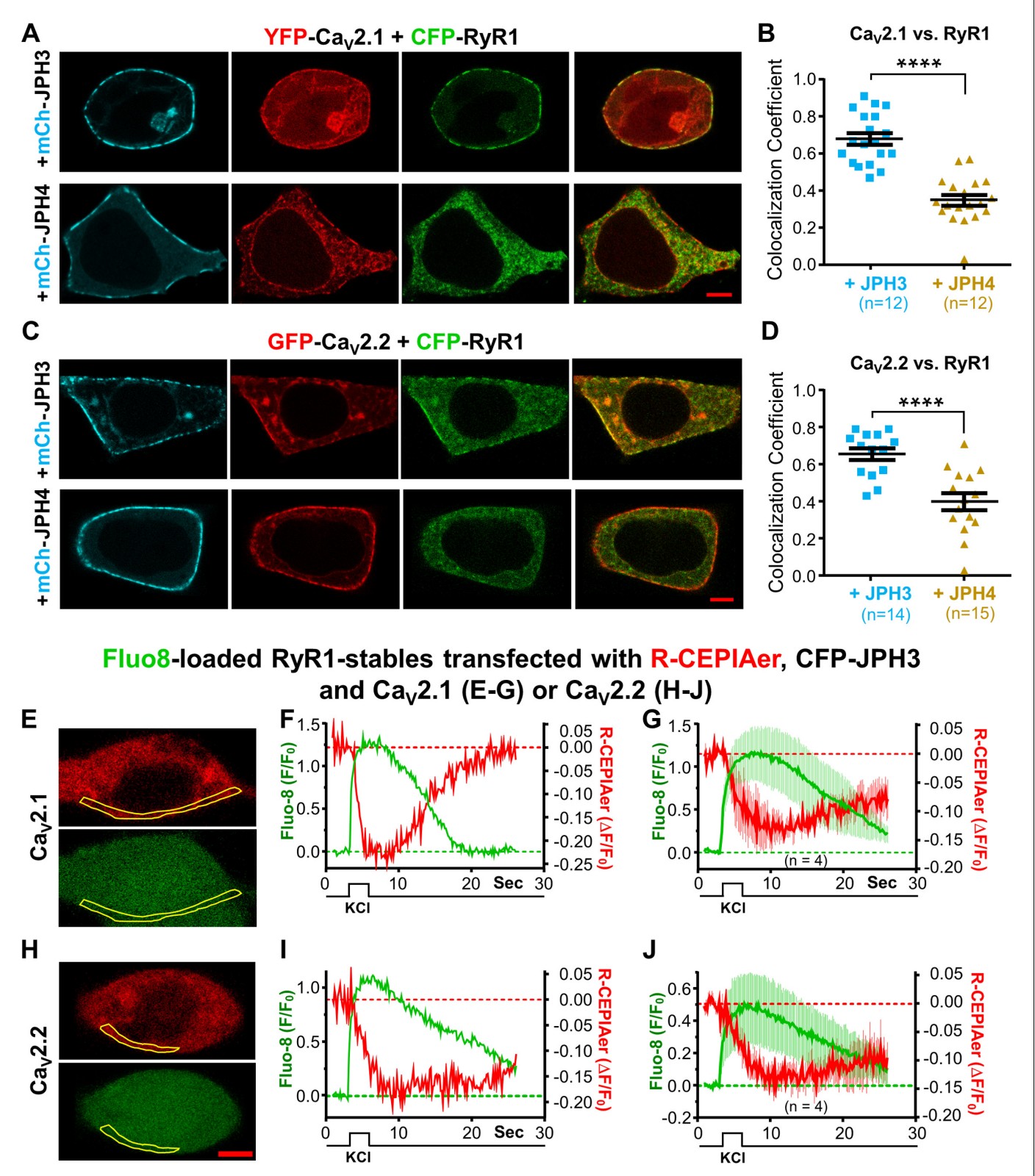

**Figure 7.** $Ca_V2.1$ and $Ca_V2.2$ colocalize with RyR1 in the presence of JPH3 but not of JPH4. $Ca^{2+}$ release from the endoplasmic reticulum (ER) is detectable after depolarization of cells expressing RyR1, JPH3, and either $Ca_V2.1$ or $Ca_V2.2$. (**A**) Representative images of cells expressing YFP-$Ca_V2.1$ (represented in red), CFP-RyR1 (represented in green), and either mCherry-JPH3 or mCherry-JPH4 (represented in cyan) as indicated. The rightmost images in each row are overlays of the YFP-$Ca_V2.1$ and CFP-RyR1 images. Pearson's coefficients for these combinations of constructs calculated from

*Figure 7 continued on next page*

*Figure 7 continued*

bottom-surface scans are plotted in (**B**). (**C**) Representative images of cells expressing YFP-Ca$_V$2.2 (represented in red), CFP-RyR1 (represented in green), and either mCherry-JPH3 or mCherry-JPH4 (represented in cyan), as indicated. The rightmost images in each row are overlays of the YFP-Ca$_V$2.2 and CFP-RyR1 images. Pearson's coefficients for these combinations of constructs, calculated from bottom-surface scans, are plotted in (**D**). In all cases (**A–D**), the cells were also transfected with β1b and α2-δ1. Statistical significance for (**B**) and (**D**): ****p $\leq$ 0.0001. (**E–J**) Cells stably transfected with RyR1 were transiently transfected with CFP-JPH3, R-CEPIAer (represented in red), β1b, α2-δ1, and either Ca$_V$2.1 (**E–G**) or Ca$_V$2.2 (**H–J**) and loaded with Fluo8-AM (represented in green). Representative images of such cells, acquired prior to depolarization, are shown in (**E**) and (**H**) for Ca$_V$2.1 and Ca$_V$2.2, respectively. (**F, I**) The Fluo8 and R-CEPIAer fluorescence for these two cells within the indicated regions of interest (outlined in yellow) is plotted as a function of time in response to a 2.5 s, focal application of 100 mM KCl. 4 of the 12 cells producing Fluo8 transients for Ca$_V$2.1 also displayed decreased ER Ca$^{2+}$. 4 of the 11 cells producing Fluo8 transients for Ca$_V$2.2 also displayed decreased ER Ca$^{2+}$. (**G, J**) Average responses of the four ER-responding cells, for each of the two construct combinations, represented as mean (solid lines) ± SEM (thin vertical lines). Pearson's coefficients plotted in (**B**) and (**D**) and their statistical comparison are provided in *Figure 7—source data 1*. Raw data for ΔF/F$_0$ plotted in (**F**), (**G**), (**I**), and (**J**) are given in *Figure 7—source data 2*.

The online version of this article includes the following source data for figure 7:

**Source data 1.** Numerical data and statistical analyses to support graphs in *Figure 7B,D*.
**Source data 2.** Numerical data to support graphs in *Figure 7F,G,I,J*.

release. The average Fluo8 and R-CEPIAer transients for these four Ca$_V$2.2-containg cells (*Figure 7J*) were similar to the transients for the Ca$_V$2.1-containing cells (*Figure 7G*). Thus, it seems that Ca$_V$2.1 and Ca$_V$2.2 may both have the ability to trigger ER calcium release via RyR1 at ER-PM junctions induced by JPH3.

## The divergent region is important for the recruitment of RyRs by JPH3

Sequence alignment indicates that the greatest divergence between JPH3 and JPH4 corresponds to a cytoplasmic region adjacent to the ER transmembrane segment (*Figure 1A*). To test the hypothesis that this divergent region is responsible for the differential recruitment of RyRs by JPH3 and JH4, we constructed a chimera ('JPH3-with-JPH4-divergent') in which the divergent region of JPH3 was replaced by the corresponding region of JPH4 (*Figure 8A*). Like JPH3 itself (*Figure 1B*), this chimera formed segmented clusters at the cell surface (*Figure 8B*, top row). However, unlike JPH3, the chimera failed to cause accumulation of either RyR1 or RyR2, and had a reduced ability to recruit RyR3 (*Figure 8B*, bottom row). Thus, the presence of the JPH4 divergent domain was sufficient to cause the chimera to behave more like JPH4 than JPH3, a pattern that was also evident in *Figure 8C*, which compares Pearson's coefficients for colocalization of the three RyR isoforms with JPH3 (left), the chimera (center), and JPH4 (right).

Because the JPH3-with-JPH4-divergent chimera displayed a loss of function with respect to accumulation of RyRs, we also attempted to test whether a gain of function would occur for the reverse chimera ('JPH4-with-JPH3-divergent'), in which the JPH4 divergent domain was replaced by that of JPH3. However, this reverse chimera failed to induce ER-PM junctions.

Although the chimera JPH3-with-JPH4-divergent lost the ability to cause effective junctional accumulation of RyRs, it appeared to retain the functions of JPH3 with respect to voltage-gated calcium channels. Specifically, a comparison of Pearson's coefficients (*Figure 8—figure supplement 1A*) indicates that the JPH3-with-JPH4-divergent chimera was effective at recruiting Ca$_V$1.2 to junctions. Moreover, this chimera slowed the inactivation of both Ca$_V$2.1 and Ca$_V$2.2 (*Figure 8—figure supplement 1B*) to an extent that was similar to that of JPH3 and not to the larger slowing caused by JPH4.

## JPH3 appears to interact with the cytoplasmic domains of RyR1 and RyR3

Because the JPH3 divergent domain is adjacent to the ER (*Figure 1A*) and is important for causing RyRs to accumulate at ER-PM junctions, it seems reasonable to hypothesize that the JPH3 divergent domain and RyR cytoplasmic domain interact with one another. To test this hypothesis, we took advantage of previous work (*Polster et al., 2018*) on a truncated RyR1 construct ('RyR1$_{1:4300}$') that encodes the bulk of the cytoplasmic domain but lacks the C-terminal regions that span the ER (SR). That work showed that RyR1$_{1:4300}$ (1) assembled into a tetrameric structure similar to that of the

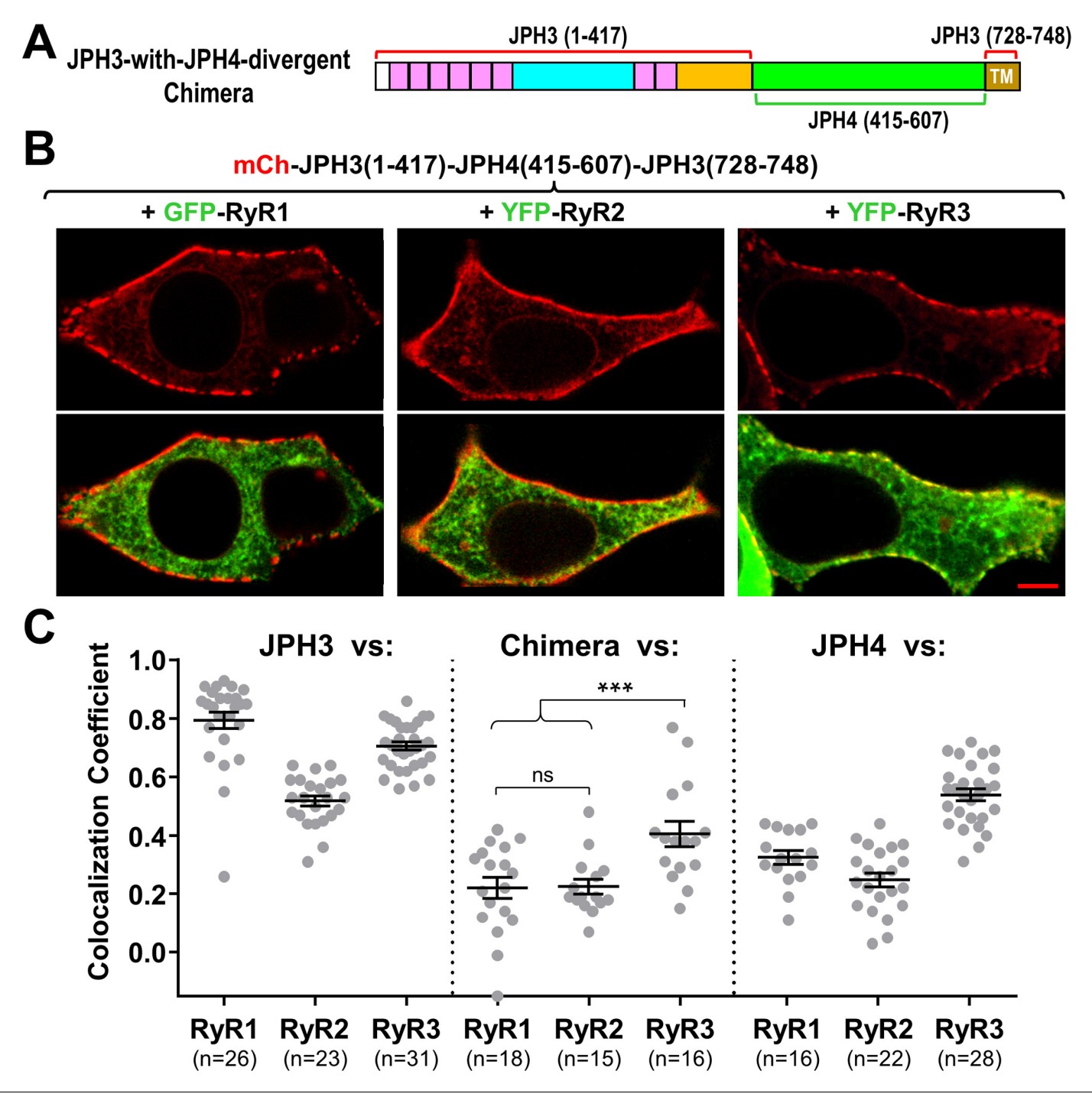

**Figure 8.** The JPH3 divergent domain is important for the junctional recruitment of all RyRs. (**A**) Schematic representation of the chimera 'JPH3-with-JPH4-divergent' in which the divergent domain of JPH3 has been replaced by that from JPH4. (**B**) Red only (top row) and red/green merged images (bottom row) of mid-level confocal sections of tsA201 cells expressing the mCherry-tagged chimera, illustrated in (**A**), together with GFP-RyR1, YFP-RyR2, or YFP-RyR3 (left to right, represented in green). Scale bar = 5 μm. (**C**) Pearson's coefficients for colocalization between the three RyR isoforms and the chimeric junctophilin (center) calculated from bottom-surface images, compared with those for JPH3 (left) and JPH4 (right), which are replotted from *Figure 5*. Statistical significance: ***p ≤ 0.001, p > 0.99 (ns). Pearson's coefficients are listed in *Figure 8—source data 1*, together with their statistical comparison to one another and to Pearson's coefficients plotted in *Figure 5*.

The online version of this article includes the following source data and figure supplement(s) for figure 8:

**Source data 1.** Numerical data and statistical analyses to support graphs in *Figure 8*.

**Figure supplement 1.** The JPH3 divergent domain is not important for interactions with voltage-gated calcium channels.

*Figure 8 continued on next page*

*Figure 8 continued*

**Figure supplement 1—source data 1.** Numerical data and statistical analyses to support graph in *Figure 8—figure supplement 1A* .
**Figure supplement 1—source data 2.** Numerical data to support graph in *Figure 8—figure supplement 1B*.

cytoplasmic domain of full-length RyR1 and (2) could bind to SR-PM junctions containing Ca$_V$1.1 but otherwise behaved as a mobile protein. Initially, we tested here whether RyR1$_{1:4300}$, and similar constructs of RyR2 ('RyR2$_{1:4226}$') and RyR3 ('RyR3$_{1:4032}$'), would localize at junctions induced by the neuronal junctophilins (*Figure 9*). In cells expressing JPH4, all the cytoplasmic domain constructs behaved as entirely mobile proteins, displaying a diffuse distribution within the cytoplasm and failing to colocalize with JPH4 at the cell surface (*Figure 9B, D–F*). A variable level of diffuse distribution was also observed for the RyR cytoplasmic domains in JPH3-expressing cells, but two of them – RyR1$_{1:4300}$ and RyR3$_{1:4032}$ – additionally displayed substantial colocalization with JPH3 at the periphery (*Figure 9A, D–F*), which was particularly evident in the bottom-surface scans. These results are consistent with the hypothesis that JPH3 interacts with the cytoplasmic domains of RyR1 and RyR3. Such an interaction would be expected to contribute to the retention of full-length RyR1 and RyR3 inserted into JPH3-containing ER-PM junctions and could help account for the observation that Pearson's colocalization coefficients for JPH3/RyR1 (~ 0.8) and JPH3/RyR3 (~ 0.7) were the highest of all the junctophilin/RyR pairs tested (*Figure 5C, E*). The failure of RyR2$_{1:4226}$ to colocalize with JPH3 (*Figure 9A, E*) and RyR3$_{1:4032}$ to colocalize with JPH4 (*Figure 9B, F*) suggests that interactions between JPH3 and the cytoplasmic domain of RyR2, and between JPH4 and the cytoplasmic domain of RyR3, are weaker and may account for the lower degree of colocalization between JPH3 and full-length RyR2, and between JPH4 and full-length RyR3 (Pearson's coefficients of ~ 0.5; *Figure 5D, E*).

Strikingly, in cells expressing both YFP-RyR3$_{1:4032}$ and mCherry-JPH3, nearly all the YFP fluorescence was concentrated at junctions and almost none was located at non-junctional regions. Moreover, such cells displayed JPH3-containing junctions that lacked colocalized RyR3$_{1:4032}$ (red-only spots, indicated by arrowheads in *Figure 9A*, bottom right). This pattern – which can be explained by a combination of (1) a strong interaction between RyR3$_{1:4032}$ and JPH3 clustered in junctions, and (2) protein levels of RyR3$_{1:4032}$ insufficient to saturate all the JPH3-containing junctions – was almost never observed for RyR1$_{1:4300}$ co-expressed with JPH3.

## Identification of a segment within JPH3 that binds the cytoplasmic domain of RyR1

As described above, we found that the junctional recruitment of full-length RyR1 and RyR3 depended on the JPH3 divergent domain in full-length junctophilin (*Figure 8*) and that the untethered cytoplasmic domains of RyR1 and RyR3 interacted with full-length JPH3 (*Figure 9*). Thus, we next tested whether the RyR cytoplasmic domains interacted with all, or part, of the JPH3 divergent domain. As a first step, we tested a truncated JPH3 construct, JPH3(1 – 707), which contains the MORN domains required for association with the plasma membrane but lacks the ER transmembrane segment necessary for inducing ER-PM junctions. *Figure 10A* illustrates a cell co-transfected with mCherry-JPH3(1 – 707) and YFP-RyR1$_{1:4300}$. Unlike full-length JPH3, which had a segmented distribution at the cell surface (*Figure 1B*), the red fluorescence of mCherry-JPH3(1 – 707) had a relatively uniform peripheral distribution (*Figure 10A*, leftmost panel). YFP-RyR1$_{1:4300}$ (represented in green in the red/green overlay of the second panel of *Figure 10A*) was both diffusely distributed in the cell interior and apparently associated with mCherry-JPH3(1 – 707), as indicated by the yellow band at the cell periphery. To probe the nature of the peripherally located YFP-RyR1$_{1:4300}$, we selectively photobleached YFP within a region of interest (ROI) inside the cell by application of 514 nm excitation applied at full-power (20 – 200-fold higher than used for imaging). Afterward, the cells were re-imaged with conditions identical to those used before bleaching with the exception that the 514 nm excitation power was doubled to provide better resolution of the reduced fluorescence of YFP-RyR1$_{1:4300}$. Red/green overlay and green-only images obtained after such a bleaching epoch are illustrated in the third and fourth panels of *Figure 10A*. Based on these images, the YFP-RyR1$_{1:4300}$ within the interior of the cell appeared to be relatively mobile since it became bleached both inside and outside the ROI. However, the YFP-RyR1$_{1:4300}$ at the cell periphery was less affected by the bleaching, consistent with the hypothesis that its mobility was reduced by an interaction with

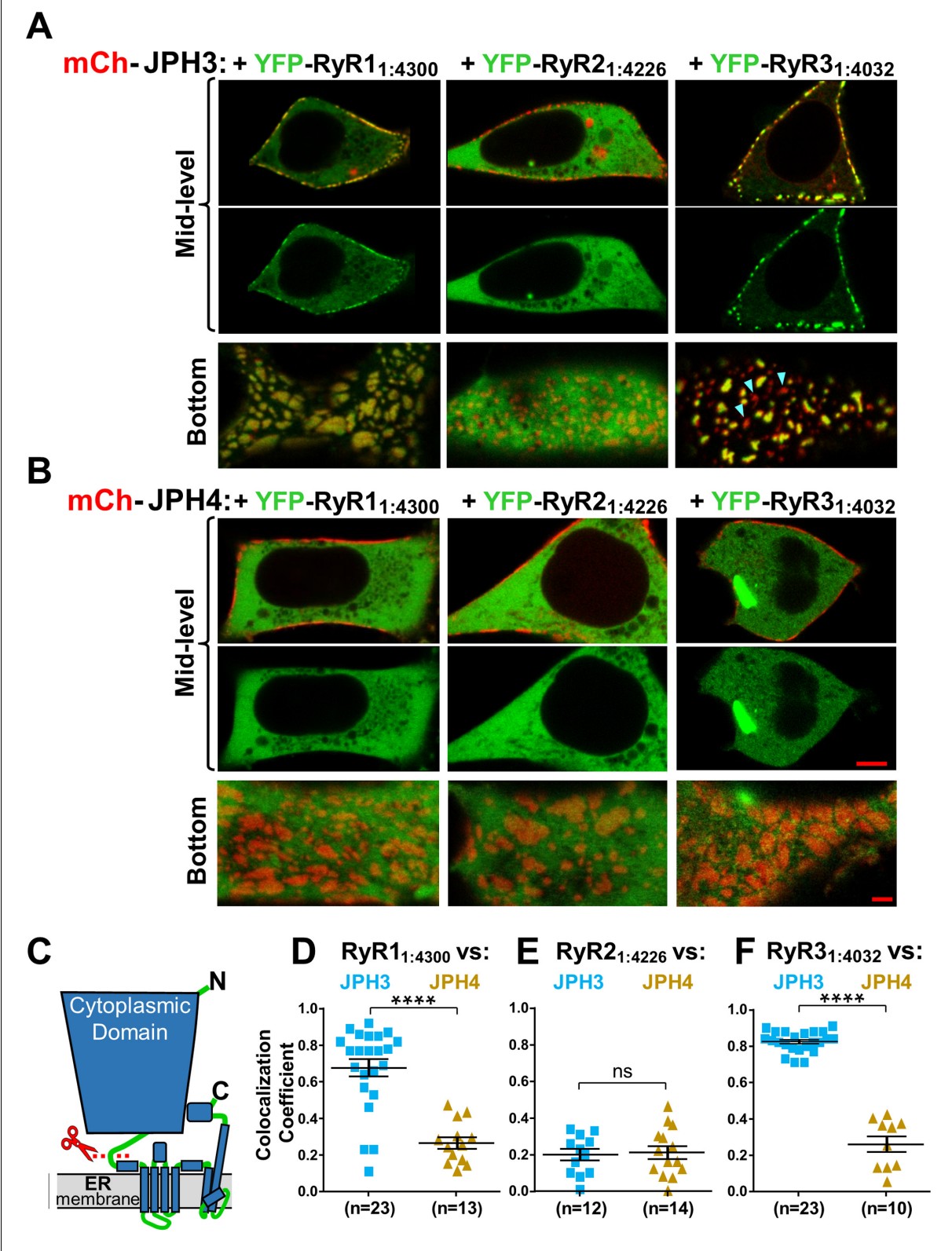

**Figure 9.** The cytoplasmic domains of RyR1 and RyR3, which have been untethered from the endoplasmic reticulum (ER), accumulate at JPH3-induced junctions between the endoplasmic reticulum and plasma membrane, but the untethered cytoplasmic domain of RyR2 does not; none of the untethered RyR cytoplasmic domains accumulate at junctions induced by JPH4. Mid-level and bottom-surface optical sections of tsA201 cells expressing YFP-RyR1$_{1:4300}$, YFP-RyR2$_{1:4226}$, or YFP-RyR3$_{1:4032}$ (represented in green) together with either mCherry-JPH3 (**A**) or mCherry-JPH4 (**B**). The

*Figure 9 continued on next page*

*Figure 9 continued*

mid-level sections are illustrated both as overlaid red/green images and green-only images (first and second rows, respectively). Note that in the presence of JPH4 YFP-RyR1$_{1:4300}$, YFP-RyR2$_{1:4226}$, and YFP-RyR3$_{1:4032}$ all behaved as large cytoplasmic proteins, which were excluded from the nucleus and lumen of the ER but were otherwise uniformly distributed. In cells co-expressing JPH3 and RyR3$_{1:4032}$, some junctions contained both proteins, whereas others contained JPH3 with little RyR3$_{1:4032}$ (indicated by arrowheads in the lower-right panel of **A**). Scale bars = 5 and 2 μm, respectively, for the mid-level and bottom-surface images. (**C**) Schematic representation of an RyR monomer, indicating the approximate position at which the large cytoplasmic domain was severed from the ER-traversing segments identified in the cryo-EM structures (*Samsó et al., 2005*; *Yuchi and Van Petegem, 2016*). (**D–F**) Pearson's colocalization coefficients for the specified construct combinations calculated from bottom-surface images. Statistical significance: ****$p < 0.0001$, $p > 0.99$ (ns). Pearson's coefficients and their statistical comparison are provided in *Figure 9—source data 1*.
The online version of this article includes the following source data for figure 9:

**Source data 1.** Numerical data and statistical analyses to support graphs in *Figure 9*.

JPH3(1 – 707). To determine whether this persistence of YFP-RyR1$_{1:4300}$ at the periphery depended on the JPH3 divergent domain, we made use of the result that YFP-RyR1$_{1:4300}$ did not interact with full-length JPH4 (*Figure 9B, D*) and constructed a chimera, JPH4(1 – 576)-JPH3(418 – 707), which consisted of JPH4 residues 1 – 576 followed by JPH3 divergent domain residues 418 – 707. *Figure 10B* illustrates images of a cell co-expressing YFP-RyR1$_{1:4300}$ and mCherry-JPH4(1 – 576)-JPH3(418 – 707), acquired both before and after photobleaching of YFP. Much like JPH3(1 – 707) itself, JPH4(1 – 576)-JPH3(418 – 707) appeared to reduce the mobility of YFP-RyR1$_{1:4300}$ at the cell surface, as might be expected if the RyR1 cytoplasmic domain and JPH3 divergent domain interacted with one another.

In order to narrow the region that might interact with the RyR1 cytoplasmic domain, we constructed cDNAs for mCherry-JPH3(418 – 748) and JPH3(653 – 748)-mCherry, which consist of all, or part, of the JPH3 divergent region linked to its ER transmembrane domain, but lack the MORN domains required for association with the plasma membrane. Cells transfected with YFP-RyR1$_{1:4300}$ together with either mCherry-JPH3(418 – 748) or JPH3(653 – 748)-mCherry are illustrated in *Figure 10C, D*. These junctophilin constructs had the reticular distribution expected for an ER localization, and YFP-RyR1$_{1:4300}$ colocalized both with mCherry-JPH3(418 – 748) and with JPH3(653 – 748)-mCherry. It was not feasible to probe this colocalization by means of photo-bleaching YFP because the cytoplasm appeared to be segmented into small compartments by these ER-associated JPH3 constructs. As an alternative, cells transfected with YFP-RyR1$_{1:4300}$ and mCherry-ER were used as a control. Little colocalization occurred between YFP-RyR1$_{1:4300}$ and mCherry-ER (*Figure 10E*). Pearson's coefficients for the various construct combinations (*Figure 10F*) indicate that the colocalization of YFP-RyR1$_{1:4300}$ differed little between JPH3(418 – 748), which contained the entire divergent region, and JPH3(653 – 748), which contained only the final approximately fourth of the divergent region. These results, taken together with the colocalization of YFP-RyR1$_{1:4300}$ with JPH4(1 – 576)-JPH3(418 – 707), as shown in *Figure 10B*, indicate that the site of interaction with the RyR1 cytoplasmic domain is contained within JPH3 residues 653 – 707. Lastly, the observation that deletion of JPH3 divergent domain residues 681 – 725 does not affect junctional recruitment of full-length RyR1 (*Figure 10—figure supplement 1*) suggests a further shortening of the candidate region to JPH3 residues 653 – 680.

As described above, RyR3$_{1:4032}$ was like RyR1$_{1:4300}$ in that it accumulated at ER-PM junctions induced by full-length JPH3 (*Figure 9*). Thus, we tested whether RyR3$_{1:4032}$ was also like RyR1$_{1:4300}$ in colocalizing with JPH3(1 – 707) at the cell surface. *Figure 11* illustrates images from a cell co-transfected with mCherry-JPH3(1 – 707) and YFP-RyR3$_{1:4032}$, which were acquired before and after photobleaching. Although there was a prominent band of red fluorescence at the cell surface, the images acquired before photobleaching show that the concentration of YFP-RyR3$_{1:4032}$ was nearly uniform throughout the cell, and images after photobleaching indicate that all of the YFP-RyR3$_{1:4032}$ was relatively mobile.

Thus, it is possible that non-identical regions of full-length JPH3 interact with YFP-RyR1$_{1:4300}$ and YFP-RyR3$_{1:4032}$, and that the region important for YFP-RyR3$_{1:4032}$ is lacking and/or altered in JPH3(1 – 707). However, we think it is also possible that JPH3(1 – 707) contains the region important for binding both RyR3$_{1:4032}$ and RyR1$_{1:4300}$. Specifically, an analysis that takes into account that the confocal acquisition parameters varied from cell to cell (see Materials and methods) indicates that the cytoplasmic YFP fluorescence intensity (mean ± SEM in arbitrary units) was 92.7 ± 22.1 (n = 14, range

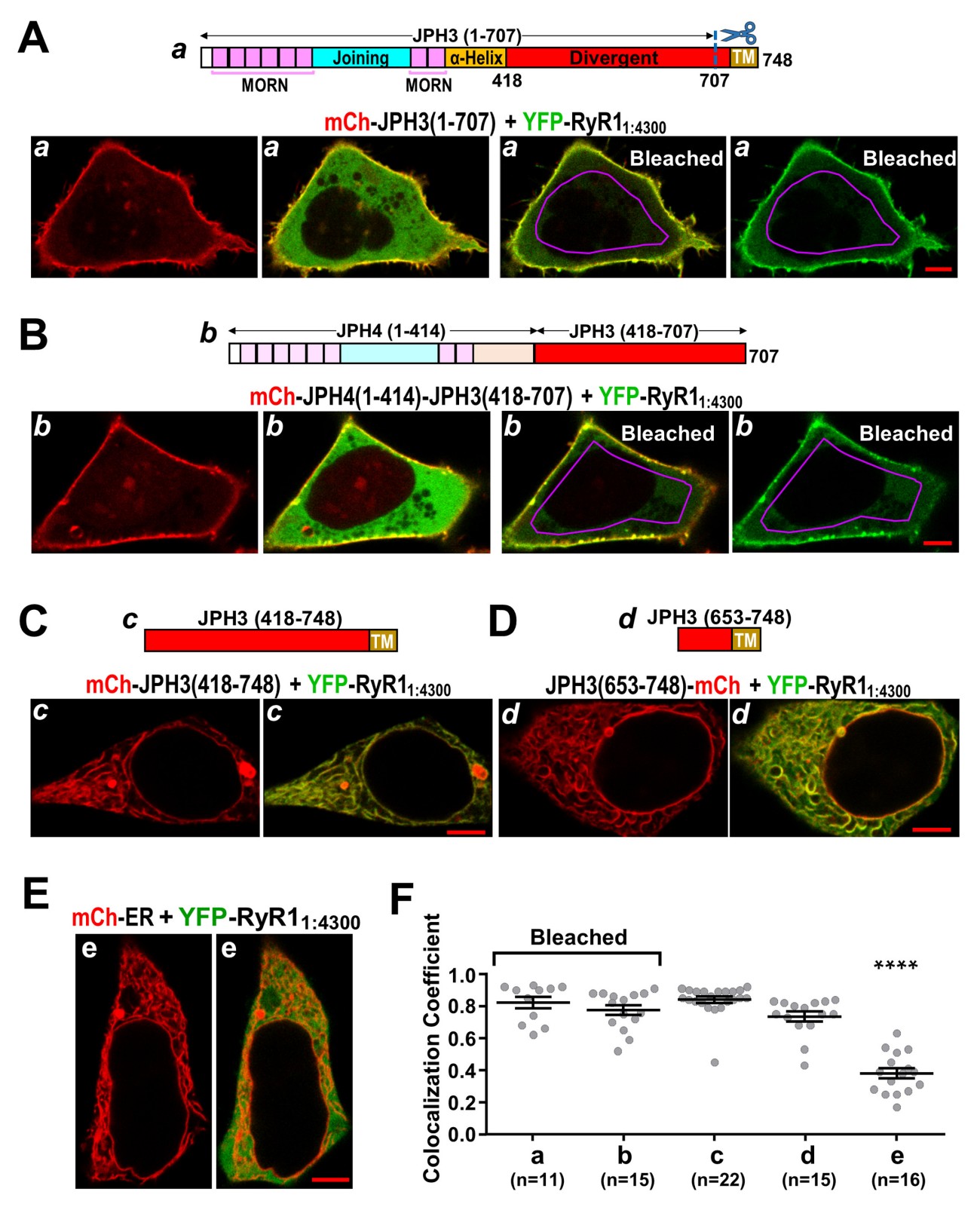

**Figure 10.** The cytoplasmic domain of RyR1 interacts with a distal segment of the JPH3 divergent region. (A–E) Confocal sections of tsA201 cells transfected with YFP-RyR1$_{1:4300}$ (represented in green) and the indicated mCherry-tagged constructs. For all the constructs, the leftmost image displays only the mCherry fluorescence (in red), and the image just to its right is a red/green overlay of the mCherry and YFP fluorescence. (A, B) The constructs JPH3(1 – 707) and JPH4(1 – 414)-JPH3(418 – 707) lack the endoplasmic reticulum (ER) transmembrane domain but have MORN motifs that cause

*Figure 10 continued on next page*

*Figure 10 continued*

association with the plasma membrane (leftmost images). The distribution of YFP-RyR1$_{1:4300}$ was similar for JPH3(1 – 707) and JPH4(1 – 414)-JPH3(418 – 707): it overlapped the junctophilin constructs at the cell surface and was also diffusely present in the cytoplasm (second images from left). After photobleaching the YFP tag within the area outlined in violet, YFP-RyR1$_{1:4300}$ remained concentrated at the cell surface as indicated both in the red/green overlays (third images from left) and in the images of only the YFP-RyR1$_{1:4300}$ fluorescence (rightmost images). See text for additional details. (C, D) mCh-JPH3(418 – 748) and JPH3(653 – 748)-mCh lack the MORN motifs required for association with the plasma membrane and were distributed in a reticular pattern in the cell interior (left), with the YFP-RyR1$_{1:4300}$ having an overlapping pattern (right). (E) YFP-RyR1$_{1:4300}$ did not colocalize with mCherry-ER. Scale bars = 5 µm. (F) Pearson colocalization coefficients for YFP-RyR1$_{1:4300}$ versus the indicated constructs, calculated from mid-level optical sections. In the case of JPH3(1 – 707) and JPH4(1 – 414)-JPH3(418 – 707), these were calculated from sections acquired after photobleaching of YFP-RyR1$_{1:4300}$ in the cell interior. **** Significantly smaller than RyR1$_{1:4300}$ versus the other four junctophilin constructs. Pearson's coefficients and their statistical comparison are provided in *Figure 10—source data 1*.

The online version of this article includes the following source data and figure supplement(s) for figure 10:

**Source data 1.** Numerical data and statistical analyses to support graph in *Figure 10*.
**Figure supplement 1.** Deletion of JPH3 divergent domain residues 681 – 725 does not affect junctional recruitment of RyR1.
**Figure supplement 1—source data 1.** Numerical data and statistical analyses to support graph in *Figure 10—figure supplement 1*.

22.4– 282.3) for YFP-RyR3$_{1:4032}$ and 1497.9 ± 317.8 (n = 16, range 80.4– 4783.4) for YFP-RyR1$_{1:4300}$. If these are assumed to be proportional to concentration, then a binding site with a $K_D$ close to the mean value for expression of YFP-RyR1$_{1:4300}$ would mean that only about ~ 15 % of JPH3(1 – 707) was occupied by even the highest level observed for YFP-RyR3$_{1:4032}$. We attempted to increase the concentration of diffusible YFP-RyR3$_{1:4032}$ by increasing the amount of cDNA used, but this appeared only to result in the formation of immobile aggregates (see *Figure 9B*, right-hand panels of rows 1 and 2, for an example).

Even with a weak binding of YFP-RyR3$_{1:4032}$ to a site in JPH3(1 – 707), substantial binding could occur to the same site in full-length JPH3 clustered at ER-PM junctions. Because of the close apposition of ER and plasma membranes at junctions, the initial capture of cytoplasmic YFP-RyR3$_{1:4032}$ by a cluster would be expected to occur at peripherally located JPH3 molecules with an affinity for YFP-RyR3$_{1:4032}$ that could be similar to that of JPH3(1 – 707). Upon unbinding from a peripheral JPH3,

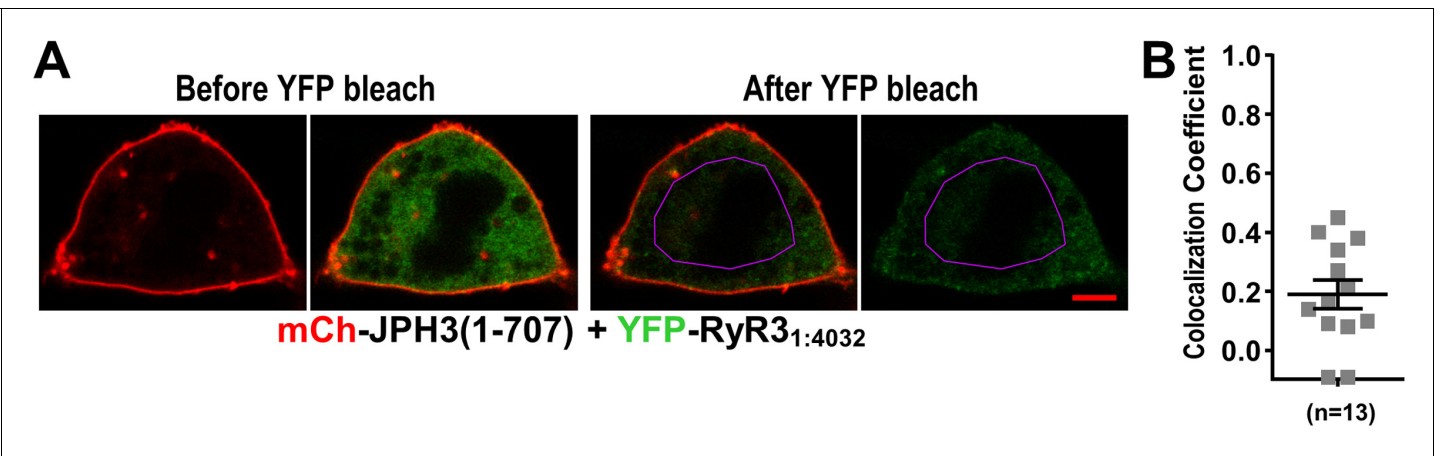

**Figure 11.** Absence of colocalization between YFP-RyR3$_{1:4032}$ and mCherry-JPH3(1 – 707) expressed in tsA201 cells. (A) Mid-level optical sections acquired from a transfected cell before and after photobleaching of YFP. mCherry-JPH3(1 – 707) was associated with the cell surface (leftmost image) but there was only weak overlap with YFP-RyR3$_{1:4032}$, which is represented in green in the red/green overlay (faint regions of yellow in the second image from left). These small regions of yellow were almost entirely absent after photobleaching of YFP within the region of interest (ROI) outlined in violet (third panel from left), indicating that they were produced by overlap between mCherry-JPH3(1 – 707) and a mobile pool of YFP-RyR3$_{1:4032}$. The rightmost panel illustrates the relatively uniform bleaching of YFP-RyR3$_{1:4032}$ both inside and outside of the ROI. Scale bar = 5 µm. (B) Pearson's colocalization coefficients for YFP-RyR3$_{1:4032}$ versus mCherry-JPH3(1 – 707) calculated from post-bleach, overlay images like that illustrated in (A). Pearson's coefficients plotted in (B) and their statistical comparison to those of RyR3$_{1:4032}$ versus full-length JPH3 (*Figure 9F*) are given in *Figure 11—source data 1*.

The online version of this article includes the following source data for figure 11:

**Source data 1.** Numerical data and statistical analyses to support graph in *Figure 11*.

YFP-RyR3$_{1:4032}$ could either return to the cytoplasm or re-bind to an adjacent JPH3, which in some cases could be closer to the center of the cluster. Because the cytoplasmic domain of RyR3 is nearly as large as the junctional gap between the ER and plasma membranes (**Protasi et al., 2000**), YFP-RyR3$_{1:4032}$ that unbound from centrally located JPH3 would differ in two important aspects from cytoplasmic YFP-RyR3$_{1:4032}$, which is free to diffuse and rotate in three dimensions. First, the diffusion of unbound YFP-RyR3$_{1:4032}$ would be largely two-dimensional within the junction. Second, the orientation necessary for interaction with JPH3 would tend to be preserved because little rotation could occur except around an axis orthogonal to the plane of the junctional membranes. Taken together, these would increase the effective concentration of YFP-RyR3$_{1:4032}$ and thus its rate of re-binding. This accentuated re-binding would slow the rate at which YFP-RyR3$_{1:4032}$ exited clusters and returned to the cytoplasm compared to the rate of unbinding from a single binding site not clustered in an ER-PM junction.

## Discussion

Here, we used colocalization of fluorescently tagged proteins expressed in tsA201 cells, together with electrophysiological measurements, to obtain insight on likely constituents of ER-PM junctions induced by the neuronal junctophilins JPH3 and JPH4. After verifying that these two proteins retained their ability to form ER-PM junctions when N-terminally tagged with fluorescent proteins (**Figure 1**), we tested three HVA Ca$^{2+}$ channels (Ca$_V$1.2, Ca$_V$2.1, and Ca$_V$2.2), and one LVA channel (Ca$_V$3.1) as potential constituents of the PM side of these junctions. The three HVA channels colocalized with both JPH3 and JPH4 (**Figure 2**), with mean Pearson's coefficients ranging from ∼ 0.5 (Ca$_V$2.1/JPH3) to ∼ 0.8 (Ca$_V$2.2/JPH4). By contrast, significantly less colocalization occurred between the LVA channel and either JPH3 or JPH4 (**Figure 2**, Pearson's coefficients < 0.4).

Based on its colocalization with JPH3 and JPH4 (**Figure 2**), Ca$_V$2.2 may be a hitherto unrecognized constituent of neuronal ER-PM junctions. Thus, it is important to take into account that Ca$_V$2.2, and also Ca$_V$2.1, may have been present in the plasma membrane at relatively low densities compared to those of Ca$_V$1.2. In particular, single-channel measurements have yielded maximum open probabilities of 0.6 for Ca$_V$2.1 (expressed in HEK293 cells; **Luvisetto et al., 2004**) and 0.5 for Ca$_V$2.2 (in frog sympathetic ganglion neurons; **Delcour and Tsien, 1993**; **Lee and Elmslie, 1999**) compared to 0.03 for Ca$_V$1.2 (in rabbit ventricular myocytes; **Lew et al., 1991**). Under the assumption that the same open probabilities are applicable to these channels co-expressed with JPH3 or JPH4, and ignoring small differences in unitary conductance, one would predict that the plasma membrane densities of Ca$_V$2.1 and Ca$_V$2.2 would be about 20- to 25- fold lower than those of Ca$_V$1.2 in order to produce the peak current densities illustrated in **Figure 3**. Thus, even though the analysis of colocalization was based on confocal scans near the surface, such scans would have included some contribution from channels that were near to, but not yet inserted into, the plasma membrane, and this could have been more significant for Ca$_V$2.1 and Ca$_V$2.2 than for Ca$_V$1.2. However, the slowing of inactivation (**Figure 3B, C**) provides evidence that JPH3 and JPH4 altered the functional environment of the majority of the Ca$_V$2.1 and Ca$_V$2.2 channels actually inserted into the plasma membrane, presumably because these channels were localized at the junctions induced by these neuronal junctophilins. Although we do not know the mechanism responsible for the slowing of inactivation, one possibility is that it involves an interaction between the junctophilins and Ca$_V$2 channels because the slowing of inactivation also occurred with truncated variants of JPH3 and JPH4, which do not induce ER-PM junctions (**Figure 4**). Whatever the exact mechanism may be, the slowing of inactivation may function to increase calcium entry via Ca$_V$2.1 and Ca$_V$2.2, and raises the possibility that JPH3 and JPH4 function not only to organize ER-PM junctions but also to modify the behavior of the signaling molecules present in those junctions.

The localization of Ca$_V$1.2 at junctions induced by JPH3 and JPH4 seems likely to depend on regions homologous to those that cause Ca$_V$1.1 to localize at triad junctions in skeletal muscle. In skeletal muscle, Ca$_V$1.1 co-IPs with both JPH1 and JPH2, an interaction for which residues 230 – 369 of JPH1 and 216 – 399 of JPH2 (human sequences) were found to be important (**Golini et al., 2011**). These sequences are reasonably well conserved for all the members of the junctophilin family, with a percentage of residue identity ranging from a minimum of 48 % (JPH1 versus JPH4) to a maximum of 66 % (JPH1 versus JPH3). In Ca$_V$1.1, a 15-residue segment within the C-terminus (amino acids 1595 – 1606) was found to bind to both JPH1 and JPH2 and to contain a motif IFFRxGGLFG that is

also present in the $Ca_V1.2$ C-terminus (*Nakada et al., 2018*). A partially conserved sequence (five identical and three conserved residues out of nine) is also found in the corresponding region of $Ca_V1.3$ (*Fujita et al., 1993*). Thus, it may be that $Ca_V1.2$ and $Ca_V1.3$ participate in interactions with JPH3 and JPH4, which are similar to those occurring between the $Ca_V1$ channels and muscle junctophilins. The search for potential sites of interaction between $Ca_V2.1$ or $Ca_V2.2$ and the neuronal junctophilins will have to proceed de novo because neither of the $Ca_V2$ C-termini contain a junctophilin-interacting motif like that in the $Ca_V1$ C-termini. For the $Ca_V2$ channels, it will also be important to determine whether a single site of interaction accounts for both selective retention at junctions and the slowing of inactivation.

Although JPH3 and JPH4 have largely overlapping abilities with respect to the recruitment of voltage-gated calcium channels in the plasma membrane (*Figures 2–4*), they differ substantially with regard to ryanodine receptors in the ER (*Figure 5*). Specifically, all three RyR isoforms colocalized with JPH3, with mean Pearson's coefficients of about 0.5 (RyR2), 0.7 (RyR3), and 0.8 (RyR1). Thus, in brain regions expressing all three RyR isoforms, RyR1 and RyR3 may be preferentially recruited to junctions containing JPH3. Examples of such regions include dentate gyrus, caudate putamen, olfactory bulb (mitral cell layer), and olfactory tubercle (*Mori et al., 2000*). By contrast with JPH3, JPH4 colocalized only with RyR3 with a mean Pearson's coefficient slightly greater than 0.5, whereas the mean coefficients were only about 0.3 for RyR1 and 0.2 for RyR2. This differential recruitment of RyRs may at least partially account for why there was no apparent behavioral phenotype observed for knockout of JPH4 only (*Moriguchi et al., 2006*), a detectable phenotype (motor discoordination) for knockout of JPH3 only (*Nishi et al., 2002*), and a broad range of neurological deficits for knockout of both JPH3 and JPH4 (*Moriguchi et al., 2006*).

We tested whether the additional expression of $Ca_V1.2$ would increase the colocalization of RyR2 with JPH3, and that of RyR3 with JPH4. No increase of colocalization occurred (*Figure 6*), and Pearson's coefficients ($\sim 0.5$) remained lower than those of either RyR1/JPH3 or RyR3/JPH3. However, the lower colocalization cannot be taken as indicating a lesser functional importance in neuronal calcium signaling. The lower colocalization could be a consequence of the presence of RyRs in both junctional and non-junctional ER. Those at the junctions would be required for the initial calcium release triggered by calcium entering across the plasma membrane, whereas those in non-junctional ER could increase the spatial spread of the cytoplasmic calcium transient. Precedent for this idea is provided by contractile cells of the heart in which RyR2 is located not only in the junctional SR, where it is activated by calcium entry via $Ca_V1.2$, but also in non-junctional ('corbular') SR (*Dolber and Sommer, 1984*; *Jewett et al., 1971*).

We also characterized the ER-PM junctions in cells transfected with YFP-tagged $Ca_V2.1$ or GFP-tagged $Ca_V2.2$ together with CFP-RyR1 and either mCherry-JPH3 or mCherry-JPH4. In the presence of JPH3, but not of JPH4, both $Ca_V2.1$ and $Ca_V2.2$ colocalized with RyR1 (*Figure 7A–D*). Furthermore, it appeared that ER calcium release could occur in cells expressing JPH3, RyR1, and either $Ca_V2.1$ or $Ca_V2.2$ (*Figure 7E–J*). More specifically, of the cells stably transfected with RyR1, and transiently transfected with $Ca_V2.1$ or $Ca_V2.2$, R-CEPIA-er and CFP-JPH3, about a third that produced cytoplasmic calcium transients in response to KCl depolarization, also showed a concomitant release of calcium from the ER. One likely contributor to this variability was variable expression of RyR1 because in cells stably transfected only with RyR1 the amplitude of caffeine transients varied substantially from cell to cell, resulting in a large standard error of the mean (*Figure 5—figure supplement 1*). Second, the cells selected for the presence of CFP-JPH3 would be expected to have had variable colocalization between the voltage-gated calcium channels and RyR1 (*Figure 7B, D*). Third, at sites of junctional contact with the plasma membrane, the depth of the ER lumen (perpendicular to the cell surface) is on the order of 100 nm or less (*Figure 1B*, bottom panels), meaning that ROIs for measurement of fluorescence (e.g., *Figure 7E, H*) would have included both junctional and non-junctional ER. Thus, a detectable change of R-CEPIA-er fluorescence would have required calcium release from both compartments. Given these limitations, the strongest statement that can be made is that the data in *Figure 7D–J* are consistent with, but do not prove, that both $Ca_V2.1$ or $Ca_V2.2$ can trigger activation of RyR1 at junctions induced by JPH3.

Evidence of binding interactions that may be important for the localization of RyR1 and RyR3 at JPH3-induced junctions is provided by the behavior of YFP-RyR1$_{1:4300}$ and YFP-RyR3$_{1:4032}$, which lack the C-terminal, pore-forming regions that anchor the full-length proteins in the ER. In particular, the fluorescence associated with YFP-RyR1$_{1:4300}$ and YFP-RyR3$_{1:4032}$ accumulated at JPH3-induced

junctions, which caused their fluorescence to be increased relative to regions lacking junctions (*Figure 9*). The accumulation of YFP-RyR1$_{1:4300}$ and YFP-RyR3$_{1:4032}$ at JPH3-induced junctions implies that these constructs bound to a component(s) present in these junctions, with JPH3 itself being an obvious candidate. If it is assumed that the binding of YFP-RyR1$_{1:4300}$ and YFP-RyR3$_{1:4032}$ indicates that they took on near-native conformations, it would imply that the cytoplasmic domains of the full-length RyR1 and RyR3 bind to JPH3 and thus help retain these RyRs at ER-PM junctions.

The failure of YFP-RyR2$_{1:4226}$ to accumulate at JPH3- or JPH4-induced junctions may have occurred because this construct failed to fold properly. For this reason, and because RyR2 has been reported to co-immunoprecipitate with JPH3 in pancreatic tissue (*Li et al., 2016*), we tested a second construct, YFP-RyR2$_{1:3991}$. This construct also failed to accumulate at junctions induced by JPH3 or JPH4 (data not shown). Possibly both YFP-RyR2$_{1:4226}$ and YFP-RyR2$_{1:3991}$ may not have folded correctly. Alternatively, it may be that the RyR2 cytoplasmic domain interacts weakly, or not at all, with JPH3 and that the co-immunoprecipitation of JPH3 and RyR2 in pancreatic tissue depends on the presence of C-terminal segments that are absent in RyR2$_{1:4226}$ and YFP-RyR2$_{1:3991}$.

Because colocalization of RyR1 with JPH3 appeared to depend on the JPH3 divergent domain (*Figure 8*), we tested for interactions between RyR1$_{1:4300}$ and constructs containing varying sized fragments of this divergent domain. We found that RyR1$_{1:4300}$ colocalized with the smallest fragment that we tested (*Figure 10D*), which consisted of JPH3 divergent domain residues 653 – 727 linked to the transmembrane domain. However, the observation that RyR1$_{1:4300}$ interacted with two different constructs containing JPH3 divergent domain residues 418 – 707 (*Figure 10A, B*) indicates that residues 708 – 727 are not required for this interaction. Additionally, full-length RyR1 colocalized with a JPH3 construct in which residues 681 – 725 had been deleted (*Figure 10—figure supplement 1*). Thus, we propose that JPH3 residues 653 – 680 contain the site that appears to be important for binding to the cytoplasmic domain of RyR1. This 28-residue segment is strongly conserved across a number of vertebrate species (*Figure 12*).

Clearly, a major goal of future investigations will be to translate our work into new examinations of ER-PM junctions in neurons, for which the results from tsA201 cells provide useful conceptual and experimental tools. For example, based on our results, we think that it will be important to determine whether P/Q currents inactivate more rapidly in cerebellar Purkinje cells lacking both JPH3 and JPH4, and whether N-type Ca$^{2+}$ channels are present in ER-PM junctions of the paraventricular nucleus of the thalamus where their transcripts are present together with those of JPH3 and JPH4. Similarly, we would like to determine whether specific functions can be assigned to RyR1 in neuronal ER-PM junctions. For this question, JPH3 constructs with altered sequence in the divergent domain may be a useful tool.

## Conclusions

In this work, we provide novel details on the role of neuronal junctophilins in the organization of ER-PM junctions. Our results suggest that JPH3 and JPH4 (1) induce ER-PM junctions and display isoform specificity in controlling the molecular architecture of the junctions and (2) serve not only to establish these calcium-signaling microdomains but also to functionally modulate at least some of

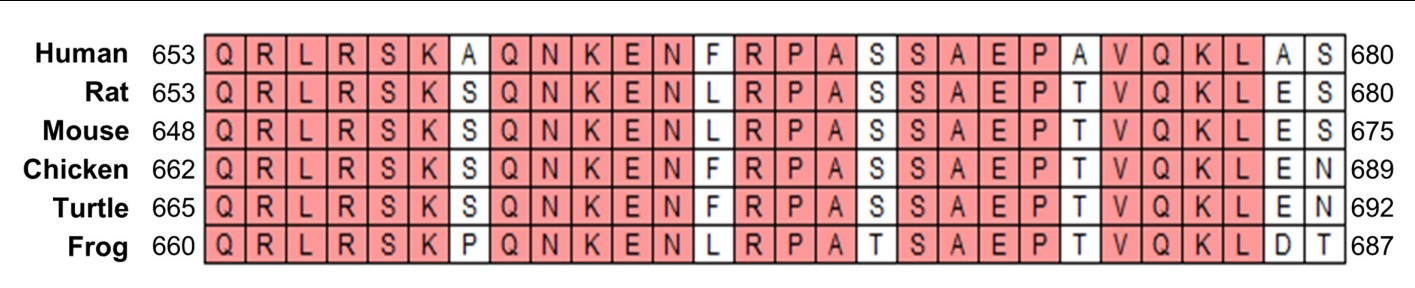

**Figure 12.** Species alignment of a segment of the JPH3 divergent domain that houses a likely site of interaction with the cytoplasmic domain of RyR1. Identical residues are shaded pink, and the numbers designate the N- and C-terminal residues, respectively. NCBI Sequence References are *AAH36533.1* (*Homo sapiens*), *NP_001100907.1* (*Rattus norvegicus*), *NP_065630.1* (*Mus musculus*), *XP_015148144.2* (*Gallus gallus*), *XP_026519458.1* (*Terrapene carolina triunguis*), and *XP_017949016.1* (*Xenopus tropicalis*).

the junctional proteins, as was directly demonstrated for $Ca_V2.1$ and $Ca_V2.2$. In addition, we have identified a 28-residue segment of JPH3 that appears to interact with the cytoplasmic domain of RyR1 and found that the JPH3 and JPH4 are modular in that there appear to be non-overlapping regions that independently interact with CaVs and RyRs.

# Materials and methods

**Key resources table**

| Reagent type (species) or resource | Designation | Source or reference | Identifiers | Additional information |
|---|---|---|---|---|
| Cell line (*Homo sapiens*) | tsA201 | tsA201 | ECACC 96121229 RRID:CVCL_0063 | 100 % STR profile match to ATCC # CRL-3216 |
| Cell line (*Homo sapiens*) | Spiking-HEK293 | PMID:24391999 | HEK293 cells stably expressing $Na_V1.3$ and $K_{IR}2.1$ | 86 % STR profile match to ATCC # CRL-1573.3 |
| Cell line (*Homo sapiens*) | RyR1-stable cells | This paper (Materials and methods) | | Spiking-HEK293 stably expressing RyR1 |
| Transfected construct (*Homo sapiens*) | JPH3 | GenScript | C96900 | In vector: pcDNA3.1-DYK with addition of mCherry or ECFP CDS |
| Transfected construct (*Homo sapiens*) | JPH4 | GenScript | C97908 | In vector: pcDNA3.1-DYK with addition of mCherry or ECFP CDS |
| Transfected construct (*Homo sapiens*) | JPH3-with-JPH4-divergent | This paper (Materials and methods) | | In vector: same as JPH3 |
| Transfected construct (*Homo sapiens*) | JPH3(1 – 707) | This paper (Materials and methods) | | In vector: pmCHerry-C1 |
| Transfected construct (*Homo sapiens*) | JPH4(1 – 576) | This paper (Materials and methods) | | In vector: pmCHerry-C1 |
| Transfected construct (*Homo sapiens*) | JPH3$_{\Delta681-725}$ | This paper (Materials and methods) | | In vector: same as JPH3 |
| Transfected construct (*Homo sapiens*) | JPH4(1 – 414)-JPH3(418 – 707) | This paper (Materials and methods) | | In vector: same as JPH3 |
| Transfected construct (*Homo sapiens*) | JPH3(418 – 748) | This paper (Materials and methods) | | In vector: pmCHerry-C1 |
| Transfected construct (*Homo sapiens*) | JPH3(653 – 748) | This paper (Materials and methods) | | In vector: pmCHerry-C1 |
| Transfected construct (*Oryctolagus cuniculus*) | $Ca_V1.2$ | PMID:2474130 | NM_001136522.1 | In vector: pEYFP-C1 or pECFP |
| Transfected construct (*Oryctolagus cuniculus*) | $Ca_V2.2$ | PMID:8386525 | GenBank: D14157.1 | In vector: modified pSP72 (see ref 28) |
| Transfected construct (*Rattus norvegicus*) | $Ca_V3.1$ | PMID:9495342 | GenBank: AF027984.1 | |
| Transfected construct (*Oryctolagus cuniculus*) | $Ca_V2.1$ | PMID:1849233 | NM_001101693.1 | In vector: pEYFP-C1 |
| Transfected construct (*Oryctolagus cuniculus*) | $\alpha2\delta1$ | PMID:28495885 | NM_001082276.1 | |
| Transfected construct (*Rattus norvegicus*) | $\beta1b$ | PMID:19996312 | GenBank: X61394.1 | |
| Transfected construct (*Oryctolagus cuniculus*) | RyR1 | PMID:2725677 | NM_001101718.1 | In vector: pEYFP-C1 or pECFP-C1 or pCEP4 (with oriP removed) |
| Transfected construct (*Mus musculus*) | RyR2 | PMID:10473538 | NM_023868.2 | In vector: pcDNA3 plus EYFP CDS |
| Transfected construct (*Oryctolagus cuniculus*) | RyR3 | PMID:12471029 | NM_001082762.1 | In vector: pcDNA3 plus EYFP CDS |

*Continued on next page*

*Continued*

| Reagent type (species) or resource | Designation | Source or reference | Identifiers | Additional information |
|---|---|---|---|---|
| Transfected construct (*Oryctolagus cuniculus*) | RyR1$_{1:4300}$ | PMID:29284662 | | In vector: pEYFP-C1 |
| Transfected construct (*Mus musculus*) | RyR2$_{1:4226}$ | This paper (Materials and methods) | | In vector: pcDNA3 plus EYFP CDS |
| Transfected construct (*Mus musculus*) | RyR2$_{1:3991}$ | This paper (Materials and methods) | | In vector: pcDNA3 plus EYFP CDS |
| Transfected construct (*Oryctolagus cuniculus*) | RyR3$_{1:4032}$ | This paper (Materials and methods) | | In vector: pcDNA3 plus EYFP CDS |
| Transfected construct (*Oryctolagus cuniculus*) | pCMV R-CEPIA1er | Addgene | Cat # 58216 RRID:Addgene_58216 | |
| Recombinant DNA reagent | pmCherry-C1 | TaKaRa/Clontech | Cat # PT3975-5 | |
| Recombinant DNA reagent | mCherry-ER | Addgene | Cat # 55041 RRID:Addgene_55041 | |
| Recombinant DNA reagent | pEYFP-C1 | TaKaRa/Clontech | Cat # 6006- 1 | |
| Recombinant DNA reagent | pECFP-C1 | TaKaRa/Clontech | Cat # 6076 -1 | |
| Recombinant DNA reagent | pCEP4 | Invitrogen | Cat # V044-50 | |
| Sequence-based reagent | #1 | This paper (Materials and methods) | PCR primer | CGGGAGCTGCCAAC CCCCTGCTGGTGGT CATGGTGATCTTGC |
| Sequence-based reagent | #2 | This paper (Materials and methods) | PCR primer | TCTAGCATGGGCTG CAGGTCTTTGGCAG TGATCCTGGCGAT |
| Sequence-based reagent | #3 | This paper (Materials and methods) | PCR primer | TCGCCAGGATCACT GCCAAAGACCTGCA GCCCATGCTAGAGG |
| Sequence-based reagent | #4 | This paper (Materials and methods) | PCR primer | AAGATCACCATGA CCACCAGCAGGG GGTTGGC |
| Sequence-based reagent | #5 | This paper (Materials and methods) | PCR primer | GCTCGCCAGTTTC TGCACG |
| Sequence-based reagent | #6 | This paper (Materials and methods) | PCR primer | CCTATCCTGGTGG TCATGGTG |
| Sequence-based reagent | #7 | This paper (Materials and methods) | PCR primer | GTACGGGCTCAGC GCCTATCGTGGTG GGAGCCGTGG |
| Sequence-based reagent | #8 | This paper (Materials and methods) | PCR primer | TGGAAGGAAGGGG AGAACTCCTGGGC TATCAGTTTGGCCA |
| Sequence-based reagent | #9 | This paper (Materials and methods) | PCR primer | TGGCCAAACTGATAG CCCAGGAGTTCTCCC CTTCCTTCCAGCACC |
| Sequence-based reagent | #10 | This paper (Materials and methods) | PCR primer | AGGGCCACGGCTCC CACCACGATAGGCG CTGAGCCCG |
| Sequence-based reagent | #11 | This paper (Materials and methods) | PCR primer | CCAGGATCACGAAT TCAGAGTTCTCCCC |
| Sequence-based reagent | #12 | This paper (Materials and methods) | PCR primer | AGTGGTACCTTCC AGGGTCAAGG |
| Sequence-based reagent | #13 | This paper (Materials and methods) | PCR primer | GAGATGAATT**C**CT TGCTGAGGATGG |
| Sequence-based reagent | #14 | This paper (Materials and methods) | PCR primer | ACGATAAGAGCA AGGGCGAGGAGG |
| Sequence-based reagent | #15 | This paper (Materials and methods) | PCR primer | CTCAGCAACACCAT GGTGGCGACC |

*Continued on next page*

*Continued*

| Reagent type (species) or resource | Designation | Source or reference | Identifiers | Additional information |
|---|---|---|---|---|
| Sequence-based reagent | #16 | This paper (Materials and methods) | PCR primer | CCATGGTGTTGCTG AGGATGGAGACGCAT |
| Sequence-based reagent | #17 | This paper (Materials and methods) | PCR primer | GCCCTTGCTCTTAT CGTCGTCATCCTTG TAATCGATGAA |
| Sequence-based reagent | #18 | This paper (Materials and methods) | PCR primer | GG**GCTAGC**GCCAC CATGCAGAGACTG CGGTCC |
| Sequence-based reagent | #19 | This paper (Materials and methods) | PCR primer | GTTCAGGGGGA GGTGTGG |
| Sequence-based reagent | #20 | This paper (Materials and methods) | PCR primer | CGTCAGATCCGCT AGCGCTACCG |
| Sequence-based reagent | #21 | This paper (Materials and methods) | PCR primer | GATCCCGGGCTA GCGGTACCGTCG |
| Sequence-based reagent | #22 | This paper (Materials and methods) | PCR primer | CCGG**G**CTAGCGGT ACCCGTCGACTGC |
| Sequence-based reagent | #23 | This paper (Materials and methods) | PCR primer | CTGATCCGATACG TGGATGAGGCGC |
| Sequence-based reagent | #24 | This paper (Materials and methods) | PCR primer | CCATCTGTTTGCCT ATGCGGCCGCTCA CCACATTACC |
| Sequence-based reagent | #25 | This paper (Materials and methods) | PCR primer | GCTCCTGCGGCCG CTCCTTCTCACTCTC |
| Commercial assay or kit | jetPRIME transfection reagent | Polyplus | VWR Cat#:89129- 922 | |
| Chemical compound, drug | Caffeine | Sigma-Aldrich | Cat# C-0750 | |
| Chemical compound, drug | Fluo8-AM | Aat Bioquest | Cat # 21082 | |
| Software, algorithm | GraphPad Prism | GraphPad Prism | RRID:SCR_002798 | Graphs and statistics |
| Software, algorithm | Fiji | ImageJ | doi: 10.1038/nmeth.2019 RRID:SCR_002285 | Image analysis |

## Expression plasmids

### JPH3 and JPH4

The cDNAs that encode human JPH3 and JPH4, with the eight-residue FLAG sequence linked to the C-terminus, were obtained from GenScript (clones C96900 and C97908, respectively). The NheI-KpnI fragment of mCherry-C1 (Takara, ref #PT3975-5, provided by Dr. M Tamkun, Colorado State University), containing the mCherry gene, was inserted at the N-term of the JPH3 and JPH4 original plasmid, cut with the same enzymes, to produce the N-terminally tagged constructs mCherry-JPH3 and mCherry-JPH4. ECFP-tagged JPH3 and JPH4 were made by cutting the mCherry coding sequence from mCherry-JPH3 and mCherry-JPH4 using the enzymes NheI and HindIII and replacing it with the ECFP encoding sequence cut with the same enzymes from the pECFP-C1 plasmid (Clontech). The JPH3-with-JPH4-divergent Chimera [(mCherry-JPH3(1 – 417)-JPH4(415 – 607)-JPH3(728 – 748))] was created from mCherry-JPH3 by replacing the sequence encoding JPH3 residues 418 – 727 with the sequence encoding JPH4 residues 415 – 607 using the Gibson assembly technique with the following primers:

#1. JPH3 Fw: CGGGAGCTGCCAACCCCCTGCTGGTGGTCATGGTGATCTTGC;
#2. JPH3 Rev: TCTAGCATGGGCTGCAGGTCTTTGGCAGTGATCCTGGCGAT;
#3. JPH4 Fw: TCGCCAGGATCACTGCCAAAGACCTGCAGCCCATGCTAGAGG;
#4. JPH4 Rev: AAGATCACCATGACCACCAGCAGGGGGTTGGC.

The proper insertion of JPH4 divergent domain was verified by sequencing. mCherry-JPH3(1 – 707) was constructed by inserting the HindIII-XbaI fragment (encoding amino acids 1 – 707) of

human JPH3 into the mCherry plasmid cut with the same enzymes. Similarly, the XhoI fragment (encoding amino acids 1 – 576) of JPH4 was inserted into the XhoI site of mCherry to produce mCherry-JPH4(1 – 576). The JPH3$_{\Delta681\text{-}725}$ construct was generated by amplifying mCherry-JPH3 construct using a forward primer starting from the codon encoding JPH3 residue 726 (#5: GC TCGCCAGTTTCTGCACG) and a reverse one starting from the codon encoding JPH3 residue 680 (#6: CCTATCCTGGTGGTCATGGTG). The ends of the resulting linear amplicon were then phosphorylated with T4 Polynucleotide Kinase (NEB, Cat#: M0201S) and allowed to re-circularize for 2 hr at room temperature in the presence of T4 ligase (NEB Cat#: M0202S). The presence of the 681 – 725 deletion was verified by sequencing.

To obtain JPH4(1 – 414)-JPH3(418 – 707), we first created a construct encoding mCherry-JPH4(1 – 414)-JPH3(418 – 727)-JPH4(608 – 628) by replacing the sequence encoding for residues JPH4 residues 415 – 607 with JPH3 residues 418 – 727 by Gibson assembly using the following primers:

> #7. JPH4 Fw: GTACGGGCTCAGCGCCTATCGTGGTGGGAGCCGTGG;
> #8. JPH4 Rev: TGGAAGGAAGGGGAGAACTCCTGGGCTATCAGTTTGGCCA;
> #9. JPH3 Fw: TGGCCAAACTGATAGCCCAGGAGTTCTCCCCTTCCTTCCAGCACC;
> #10. JPH3 Rev: AGGGCCACGGCTCCCACCACGATAGGCGCTGAGCCCG.

The proper insertion of JPH3 residues 418 – 707 was verified by sequencing, but the construct did not induce ER-PM junctions. The JPH4(1 – 414)-JPH3(418 – 707) construct was made by digesting the (non-functional) mCherry-JPH4(1 – 414)-JPH3(418 – 727)-JPH4(608 – 628) with HindIII and XbaI and pasting the cut fragment, encoding amino acids JPH4(1 – 414)-JPH3(418 – 707), into the mCherry empty vector cut with the same enzymes. To create mCherry-JPH3(418 – 748), a nucleotide sequence, starting with the codon encoding residue 418 and ending 96 bp after the stop codon of the FLAG tag fragment, was amplified using the following primers designed to add restriction sites (indicated in bold) for EcoRI at the 5′ and KpnI at the 3′ ends respectively:

> #11. EcoRI Fw: CCAGGATCACGAATTCAGAGTTCTCCCC;
> #12. KpnI Rev: AGTGGTACCTTCCAGGGTCAAGG.

EcoRI and KpnI enzymes were then used to cut and paste the amplicon into mCherry empty vector. JPH3(653 – 748)-mCherry was built from a previously created mCherry-JPH3(617 – 748), which was obtained similarly to the mCherry-JP3(418 – 748) described above, using the same reverse 'KpnI Rev' primer and a new primer designed to amplify a sequence starting from residue 617 instead of 418, and add a restriction site (indicated in bold) for EcoRI (#13: EcoRI Fw 617: GAGATGAATTCC TTGCTGAGGATGG). The mCherry tag of the JPH3(617 – 748) construct was then switched from the N-terminus to the C-terminus of the JPH3 fragment using Gibson assembly. To do so, we amplified a sequence starting with the codon encoding residue JPH3 617 and ending immediately before the stop codon of the FLAG tag. The amplified sequence was then inserted right after the Kozak sequence (including the ATG codon) of the mCherry tag in the mCherry-C1 empty vector. The primers used were:

> #14. mCherry Fw: ACGATAAGAGCAAGGGCGAGGAGG;
> #15. mCherry Rev: CTCAGCAACACCATGGTGGCGACC;
> #16. mCh-JPH3(617 – 748) Fw: CCATGGTGTTGCTGAGGATGGAGACGCAT;
> #17. mCh-JPH3(617 – 748) Rv: GCCCTTGCTCTTATCGTCGTCATCCTTGTAATCGATGAA.

Finally, part of the JPH3(617 – 748)-mCherry construct was amplified using a Fw primer (#18: GGGCTAGCGCCACCATGCAGAGACTGCGGTCC) designed to add a NheI restriction site (indicated in bold) and the Kozak sequence (including the ATG codon) upstream to residue 653, and a Rev primer (#19: GTTCAGGGGGAGGTGTGG) designed to include the multiple cloning site, already present at the 3′ of the mCherry tag, in the amplicon. NheI and HindIII enzymes were then used to replace the 617-mCherry segment of the JPH3(617 – 748)-mCherry construct, with the new, shorter amplicon to generate JPH3(653 – 748)-mCherry.

## mCherry-ER

The expression plasmid for mCherry-ER-3 was obtained from Addgene (Cat # 55041). Because of the presence of calreticulin signal peptide and ER retention KDEL sequences, respectively at the

N-term and C-term of mCherry, the mCherry-ER construct is optimized to function as a luminal ER marker.

### R-CEPIAer

The plasmid, encoding an ER lumen-targeted calcium indicator protein (*Suzuki et al., 2014*), was obtained from Addgene (Addgene plasmid # 58216).

### Voltage-gated channels and sub-units

The construction of EYFP-Ca$_V$1.2, ECFP-Ca$_V$1.2, and EGFP-Ca$_V$2.2 (channels having rabbit sequence) was described previously (*Polster et al., 2015*; *Grabner et al., 1998*). Rat Ca$_V$3.1-EYFP (*Fang and Colecraft, 2011*) was kindly provided by Dr. H. Colecraft (Columbia University, NY). To produce EYFP-Ca$_V$2.1, the Ca$_V$2.1 coding sequence was excised with HpaI and SalI from EGFP-Ca$_V$2.1 (*Grabner et al., 1998*) and ligated into pEYFP-C1 (Clontech) that had been cut with the same two enzymes. Unlabeled rabbit $\alpha_2$-$\delta_1$ (Sequence ID: NM_001082276.1) was kindly provided by Dr. W. A. Sather (University of Colorado). To produce unlabeled rat $\beta$1b, its coding sequence was excised from ECFP-$\beta$1b (Sequence ID: NM_017346, kindly provided by Dr. S. Papadopoulos, University of Cologne) with HindIII and KpnI and inserted in place of the Ca$_V$1.1 gene in the 'unlabeled $\alpha$1s' plasmid, previously constructed by *Papadopoulos et al., 2004*.

### RyR constructs

The rabbit RyR1 construct N-terminally tagged with GFP was described by *Lorenzon et al., 2001*. The construction of the N-terminally labeled constructs EYFP-RyR1, ECFP-RyR1, and RyR1$_{1:4300}$ (in which the coding sequence for RyR1 terminates at amino acid 4300) was also described previously (*Polster et al., 2018*). Unlabeled RyR1, used for generating stable cell lines, was created by cutting the RyR1 sequence of EYFP-RyR1 with HindIII and MfeI and ligating it into the pCEP4 plasmid cut with the same two enzymes. Note that pCEP4 was originally designed for extrachromosomal replication. In our case, the digestion with HindIII and MfeI removes the oriP sequence from the vector backbone, eliminating the possibility of extrachromosomal replication. EYFP-RyR2 and EYFP-RyR3 were created by PCR amplification of the EYFP sequence in pEYFP-C1 using the following primers designed to add an additional NheI cutting site at the 3′ end of the EYFP gene (indicated in bold) and to shift the EYFP gene reading frame to match that of RyR2 and RyR3.

> #20. Fw primer: CGTCAGATCCGCTAGCGCTACCG (used for both EYFP-RyR2 and RyR3);
> #21. Rev primers: GATCCCGGGCTAGCGGTACCGTCG (used for EYFP-RyR2) and
> #22. CCGGGCTAGCGGTACCCCGTCGACTGC (used for EYFP-RyR3).

The EYFP coding sequence excised with NheI was inserted at the RyR N-terminal in the expression plasmids for mouse RyR2 (*Zhao et al., 1999*) and mouse RyR3 (*Jiang et al., 2003*) (kindly provided by Dr. W. Chen, University of Calgary), which had been cut with the same enzyme. The construction of the truncated EYFP-RyR2 made use of unique restriction sites present in the EYFP-RyR2 plasmid: BsiWI (Pro2995 of RyR2) and NotI (3′ to the stop codon). RyR2$_{1:3991}$ and RyR2$_{1:4226}$ were obtained by designing primers to amplify and create a new NotI site (indicated in bold) at the 3′ of the C-terminal regions extending from the BsiWI restriction site to residues Val3991 or Pro4226.

> #23. Fw primer: CTGATCCGATACGTGGATGAGGCGC;
> #24. Rev primers: CCATCTGTTTGCCTATGCGGCCGCTCACCACATTACC (for RyR2$_{1:3991}$) and
> #25. GCTCCTGCGGCCGCTCCTTCTCACTCTC (for RyR2$_{1:4226}$).

These PCR fragments were digested with BsiWI and NotI and inserted into the EYFP-RyR2 plasmid that had been cut with the same enzymes. All the amplified constructs were verified by DNA sequencing to exclude the presence of mutations introduced by the polymerase. Truncated EYFP-RyR3 (RyR3$_{1:4032}$) was created by digesting the EYFP-RyR3 plasmid with the double-cutting AvrII, isolating the plasmid from the cut fragment (the terminal part of the C-term) and allowing it to re-circularize.

## Cell culture and cDNA transfection

tsA201 cells (100 % STR profile match to HEK293T, ATCC Cat # CRL-3216) were cultured in high-glucose Dulbecco's Modified Eagle Medium (Mediatech), supplemented with 10 % (vol/vol) FBS and 2 mM glutamine in a humidified incubator with 5 % (vol/vol) $CO_2$. Spiking-HEK cells (86 % STR profile match to HEK293, ATCC Cat # CRL-1573.3), provided by Dr. Adam Cohen, Harvard University, were cultured in the same medium as described for tsA201 cells with the addition of 2 μg/ml puromycin and 500 μg/ml geneticin (G418). For culturing RyR1-stable cells (spiking-HEK stably transfected with RyR1), an additional 300 μg/ml of hygromycin was added to the spiking-HEK medium. The tsA201 and RyR1-stable cells were tested by the Tissue Culture Core at the University of Colorado Anschutz Medical Campus and found to be negative for mycoplasma.

Cells at ≈ 70 % confluence were transfected by exposure for 3.5 hr to the jetPRIME reagent (Polyplus-transfection Inc, NY) containing either 1 μg ($Ca_V$, RyR constructs, and R-CEPIAer) or 0.5 μg (β1b, α2δ1, and junctophilin constructs) per 35 mm plastic culture dish (Falcon). After 3.5 hr of transfection, the cells were rinsed and either maintained overnight in fresh medium in the same dish for electron microscopy or detached from the dish using Trypsin-EDTA (Mediatech) and replated at ~ $1.5 \times 10^4$ cells/dish in 35 mm plastic culture dishes for electrophysiology or at ~ $2.5 \times 10^4$/cm$^2$ in glass-bottomed microwell dishes (MatTek, 35 mm dish, 14 mm microwell diameter), previously coated with collagen type III (Sigma-Aldrich) or ECL (Millipore), for confocal imaging.

## Generation of RyR1-stable cells

Spiking-HEK293 cells were transfected (see above) with RyR1-pCEP4 and propagated in spiking-HEK medium (described above) supplemented with 300 μg/ml hygromycin B (Invitrogen) for selection. After establishing a hygromycin-resistant polyclonal culture, the cells were plated at low density (≈ 200 cells/10 cm dish) and maintained for several days until the isolated single cells had expanded into monoclonal colonies of about 50 cells or more. The cells were then loaded with a calcium indicator (Fluo3-AM, Thermo Fisher) and tested for their response to localized application of 1 mM caffeine. The colony showing the highest, most uniform response to caffeine was isolated, subcultured, and expanded to be used for experiments.

## Electron microscopy

Twenty-four hours after transfection, cells were detached with Trypsin-EDTA, pelleted, and fixed with 5 % glutaraldehyde in 0.1 M sodium cacodylate buffer (pH = 7.4). The pellets were postfixed in 2 % (vol/vol) $OsO_4$ in 0.1 M cacodylate buffer for 1 hr at 4 ℃, enbloc-stained with saturated uranyl acetate in $H_2O$, embedded in EPON and sectioned. The sections were post-stained with lead citrate (*Hanaichi et al., 1986*) before imaging with a FEI Tecnai transmission electron microscope.

## Electrophysiology

All experiments were performed at room temperature (~ 25 ℃). Pipettes were fabricated from borosilicate glass and had resistances of ~ 2.5 MΩ when filled with an internal solution consisting of (in mM): 140 Cs-aspartate, 10 Cs-EGTA, 5 $MgCl_2$, and 10 HEPES (pH 7.4, with CsOH). The bath solution contained (mM) 145 tetraethylammonium-Cl (TEA-Cl), 10 $CaCl_2$ (or 10 $BaCl_2$ where indicated), and 10 HEPES (pH 7.4 with TEA-OH). To record $Ca^{2+}$ currents, cells were held at a potential of − 60 mV (− 70 mV for $Ca_V$1.3) and then depolarized to potentials ranging from − 20 to +70 mV (− 40 to + 70 mV for $Ca_V$1.3). Electronic compensation was used to reduce the effective series resistance to < 8 MΩ (time constant < 500 μs). Linear components of leak and capacitive current were corrected with −P/4 online subtraction protocols. Filtering was set at 1 – 2 kHz and digitization at 20 kHz. Channel inactivation was quantified as the percentage of peak current remaining 700 ms after the peak ($I_{700}$/$I_{peak}$).

## Live-cell calcium imaging

### Caffeine transients

Untransfected tsA201 cells and RyR1-stable cells were cultured on glass-bottomed dishes and loaded with Fluo8-AM in serum-free medium for 10 min at 37 ℃. After loading, cells were superfused with rodent ringer containing (in mM): 146 NaCl; 5 KCl; 2 $CaCl_2$; 1 $MgCl_2$; 10 HEPES (pH 7.4 with NaOH). Individual cells were then stimulated by focal application of 1 mM caffeine (dissolved in

rodent ringer) over the cell for 1.5 s using a Picospritzer. Fluo8 was imaged at 4 frames/s (250 ms/frame).

## KCl stimulation

RyR1-stable cells were transfected with CFP-JPH3, β1b, α2δ1, R-CEPIAer, and either YFP-Ca$_V$2.1 or GFP-Ca$_V$2.2 as described above. Transfected cells, cultured on glass-bottomed dishes, were then loaded with Fluo8-AM as described above and then superfused with rodent ringer solution (composition reported above). Transients were triggered using a Picospritzer to apply 100 mM KCl rodent ringer (K$^+$ replacing Na$^+$) for 2.5 s on expressing cells. Candidate expressing cells were chosen by the presence of clear CFP-JPH3 junctions at the periphery of the cell. Fluo8 transients and R-CEPIAer transients were imaged simultaneously at 6.6 frames/s (150 ms/frame).

## Imaging

Cells were superfused with physiological saline (in mM: 146 NaCl, 5 KCl, 2 CaCl$_2$, 1 MgCl$_2$, 10 HEPES, pH 7.4, with NaOH) and imaged using a Zeiss 710 confocal microscope. Images were obtained as single optical sections (~ 0.9 µm thick) with a 63× (1.4 NA) oil immersion objective. Fluorescence excitation (Ex) and emission (Em) (nanometers) were CFP (Ex, 440; Em, 454 – 508), GFP (Ex, 488; Em, 493 – 586), YFP (Ex, 514; Em, 515 – 619), mCherry (Ex, 543; Em, 578 – 696), Fluo8 (Ex, 488; Em 493 – 570), and R-CEPIAer (Ex, 543; Em 593 – 677). The analysis of colocalization was carried out on ~ 0.9 µm optical sections acquired at the bottom surface of the cell, close to the glass substrate. Cells were chosen for analysis solely by the presence of distinguishable surface foci of junctophilin, regardless of the fluorescence distribution of the co-expressed proteins. Fluorescence profiles were obtained using the 'profile function' in the Zeiss Zen Black software. Pearson's colocalization coefficients were calculated semi-automatically with ImageJ using a custom macro designed to *Porter and Palade, 1957* perform background subtraction and median filtering (2-pixel radius), and (*Rosenbluth, 1962*) run the ImageJ 'Coloc2' plugin that calculated the above-threshold Pearson's coefficient with the following settings: threshold regression type = bisection, PSF = 10, Costes' randomizations = 10. For cells expressing JPH constructs that formed ER-PM junctions, Pearson's coefficients were calculated from ~ 0.9 -µm-thick optical sections at the cell's substrate-adhering surface. For samples expressing JPH fragments that associated only with the internal ER or only with the plasma membrane, Pearson's coefficients were calculated from ~ 0.9 -µm-thick optical sections acquired roughly halfway between the bottom and top surfaces of the cell, excluding the nucleus and obvious protein aggregates (rarely present) from the analysis. For experiments in which photobleaching was employed, Pearson's coefficient was calculated on images acquired immediately after 10 photobleaching scans (100 % of 514 nm laser power) applied to a ROI within the cell interior. Each post-bleaching image was acquired at twice the laser power used to acquire the corresponding pre-bleached image. The resulting Pearson's coefficients for each cell were plotted in a dot plot, together with the mean ± SEM for the entire group of cells.

To estimate the relative levels at which YFP-RyR1$_{1:4300}$ and YFP-RyR3$_{1:4032}$ could accumulate in the cytoplasm, we used cells in which they had been co-expressed with mCherry-JPH4 because (1) neither construct appeared to interact with JPH4 (see *Figure 7B*) and (2) the cells had been selected only based on mCherry fluorescence. Because the confocal acquisition parameters varied from cell to cell (to maximize the dynamic range of the images), the measured YFP fluorescence intensities were corrected by making measurements with these sets of acquisition parameters applied to individual cells expressing only YFP.

## Statistical methods

Student's t-test with Welch's correction was used for comparison between two sets of data. One-way ANOVA, with Tukey's post-hoc test was performed for comparison of multiple sets of data. A detailed description of the results of all statistical analyses is reported in the 'Raw data and statistics'.

## Acknowledgements

The cDNA for Ca$_V$3.1 was provided by Dr. Henry Colecraft. The cDNAs for RyR2 and RyR3 were provided by Dr. Wayne Chen. The spiking HEK293 cells were provided by Dr. Adam Cohen.

## Additional information

### Competing interests

Kurt Beam: Reviewing editor, *eLife*. The other author declares that no competing interests exist.

### Funding

| Funder | Grant reference number | Author |
| --- | --- | --- |
| NIH Office of the Director | R01 AR070298 | Kurt Beam |

The funders had no role in study design, data collection and interpretation, or the decision to submit the work for publication.

### Author contributions

Stefano Perni, Conceptualization, Formal analysis, Investigation, Methodology, Writing - review and editing; Kurt Beam, Conceptualization, Resources, Supervision, Funding acquisition, Writing - original draft

### Author ORCIDs

Stefano Perni https://orcid.org/0000-0002-0591-4376
Kurt Beam https://orcid.org/0000-0001-6902-085X

### Decision letter and Author response

Decision letter https://doi.org/10.7554/eLife.64249.sa1
Author response https://doi.org/10.7554/eLife.64249.sa2

## Additional files

### Supplementary files

- Transparent reporting form

### Data availability

Raw data for peak current vs voltage, inactivation vs voltage, and Pearson's coefficients have been provided with the uploaded manuscript files.

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
