## [Decision Letter]

**Acceptance summary:**

This manuscript examines the interplay between transiently expressed neuronal junctophilins 3 and 4, voltage-dependent calcium (Ca_v_2) channels, and ryanodine receptors. The authors find that both junctophilin isoforms alter the gating behavior of Ca_v_2.1 and Ca_v_2.2 channels. They also show that junctophilin 3 is very effective in recruiting ryanodine receptors to the endoplasmic reticulum-plasma membrane junction, whereas junctophilin 4 is less effective.

**Decision letter after peer review:**

Thank you for submitting your article "Neuronal Junctophilins Recruit Specific Ca_V_ and RyR isoforms to ER-PM Junctions and Functionally Alter Ca_V_2.1 and Ca_V_2.2" for consideration by *eLife*. Your article has been reviewed by 3 peer reviewers, and the evaluation has been overseen by a Reviewing Editor and Kenton Swartz as the Senior Editor. The following individuals involved in review of your submission have agreed to reveal their identity: Gerald Zamponi (Reviewer #1); Mitsuhiko Yamada (Reviewer #2).

The reviewers have discussed the reviews with one another and the Reviewing Editor has drafted this decision to help you prepare a revised submission.

Summary:

This exceptionally well written manuscript examines the interplay between transiently expressed junctophilins 3 and 4, Ca_v_2 channels, and ryanodine receptors. The authors find that both junctophilin isoforms alter the gating behavior of Ca_v_2.1 and Ca_v_2.2 channels. They also show that junctophilin 3 is very effective in recruiting Ryanodine receptors to the ER-plasma membrane junction, whereas junctophilin 4 is less effective. In some ways, this tracks well with the notion that knockout of junctophilin 3 has a much more severe phenotype than that of junctophilin 4.

Essential revisions:

1. The manuscript really has two foci – one is on calcium channels, and the other is on recruitment of Ryanodine receptors. The latter is explored in much greater detail and could easily form the basis for a free standing paper. On the other hand, one experiment that would bring the two aspects together would be to see whether the recruitment of RyRs is modulated by the presence of a calcium channel (either Ca_v_2.2 or Ca_v_2.1), and perhaps vice versa. Along these lines, does the presence of an overexpressed RyR alter the effect of junctophilin on inactivation of Ca_v_2? Presumably tsa cells have endogenous RyRs, so what if the more pronounced effect of junctophilin 4 on Ca_v_2 channels is due to a reduced ability of this junctophilin isoform to recruit RyRs to the complex… either one of these suggested experiments would be quite easy to do if the authors so desired.

2. In the 'artificial junctions' in tsA201 cells, are there any functional interactions between Ca_V_ and RyR (i.e. CICR or CDI) or between JP3 and JP4. This sort of matters may be important in order to infer the functional importance of neuronal ER-PM junctions from this reconstitution study.

3. The finding that Jph3 and Jph4 selectively slows-down inactivation of Ca_V_2 channels but not Ca_V_1 is novel and highly interesting. For these studies, the authors use Ca^2+^ as charge carrier with high intracellular Ca^2+^ buffering (10 mM EGTA). As Ca^2+^-dependent inactivation (CDI) of Ca_V_2 channels requires global Ca^2+^ elevation, this change in inactivation of Ca_V_2.1 and Ca_V_2.2 likely reflects a change in voltage-dependent inactivation (VDI) as the authors point out. However, CDI of Ca_V_1.2 channels is sensitive to local Ca^2+^ which may be quite strong under these conditions. If JPH were to only affect VDI, then strong CDI could potentially mask underlying differences in VDI of Ca_V_1.2. Have the authors considered using Ba2+ current measurements to further establish the absence of any effect of JPH on VDI of Ca_V_1.2?

4. Comparing figures 3 and 4, full length JPH3 only modestly slows inactivation of Ca_V_2.1. However, the truncated variant (1-707) exhibits stronger inhibition of inactivation. What causes this difference? Could it be be because JPH3 only partially recruits Ca_V_2.1 to the ER-PM junctions, while the truncated form can act on channels that may be outside of the junctional space? The difference in potency appears to be lesser for Ca_V_2.2, which has a higher colocalization with JPH3 than Ca_V_2.1

5. Statistical analysis comparing colocalization coefficients would be helpful, particularly in inferring rank order of JPH3 preference with different RyR isoforms (line 259). Given the variability in the colocalization coefficients, it is possible that some of the apparent differences may not be statistically significant. For e.g. JPH3:RyR1 versus JPH3:RyR3 or JPH4:RyR2 versus JPH4:RyR1.

6. One shortcoming of the present study is that the differential association of JPH/RyR/Ca_V_ channel complexes are probed only in TSA cells under overexpression conditions. While this allows for comparison of the propensity of JPH to recruit different ion channels, it may not reflect the composition of native ER-PM junctions in neurons. This study may be further bolstered by analysis of co-localization of endogenous RyR, JPH, and Ca_V_ channels in neurons.

---

## [Author Response]

Essential revisions:1. The manuscript really has two foci – one is on calcium channels, and the other is on recruitment of Ryanodine receptors. The latter is explored in much greater detail and could easily form the basis for a free standing paper. On the other hand, one experiment that would bring the two aspects together would be to see whether the recruitment of RyRs is modulated by the presence of a calcium channel (either Ca_v_2.2 or Ca_v_2.1), and perhaps vice versa.

In the revised manuscript, we have attempted to bring the two aspects together by examining interactions between the Ca_V_s and RyRs as described here and in our response to Point 2. Specifically, we have added results showing that Ca_V_2.1 colocalizes with RyR1 when JPH3 is also present but not with JPH4 (Figure 7A, B). Similarly, JPH3, but not JPH4, jointly recruits Ca_V_2.2 and RyR1 (Figure 7C, D). Another set of new experiments tested whether the additional presence of Ca_V_1.2 affects the recruitment of RyR2 by JPH3 or the recruitment of RyR3 by JPH4 (new Figure 6).

Along these lines, does the presence of an overexpressed RyR alter the effect of junctophilin on inactivation of Ca_v_2? Presumably tsa cells have endogenous RyRs, so what if the more pronounced effect of junctophilin 4 on Ca_v_2 channels is due to a reduced ability of this junctophilin isoform to recruit RyRs to the complex… either one of these suggested experiments would be quite easy to do if the authors so desired.

We did not test whether overexpressed RyRs altered the effect of junctophilin on the inactivation of Ca_V_2. However, we do present results in the revised manuscript on calcium transients in response to caffeine as an indication of endogenous expression of RyR in the tsA201 cells (Figure 5-extended figure 1). Caffeine transients were undetectable in the control cells. Of course, this does not exclude the possibility that JPH3 is more effective at junctionally recruiting the small number of endogenous RyRs than is JPH4. However, this does not seem sufficient to explain why JPH4 has a larger effect on inactivation of Ca_v_2 channels than JPH3 because the chimera JPH3-with-JPH4-divergent is less effective at recruiting exogenous RyRs than JPH4 (Figure 8), but its effect on inactivation is like that of JPH3 and not like that of JPH4 (Figure 8—figure supplement 1).

2. In the 'artificial junctions' in tsA201 cells, are there any functional interactions between Ca_V_ and RyR (i.e. CICR or CDI) or between JP3 and JP4. This sort of matters may be important in order to infer the functional importance of neuronal ER-PM junctions from this reconstitution study.

We have added results testing for CICR in cells transfected with Ca_V_2.1/RyR1/JPH3 and in cells transfected with Ca_V_2.2/RyR1/JPH3. For these experiments it was necessary to directly monitor intra-ER Ca^2+^ (we used R-CEPIAer) since entry of extracellular Ca^2+^ via the Ca_v_2 channels would produce a cytoplasmic calcium transient whether or not CICR occurred. For reasons discussed further in the manuscript, the measurements were carried out on "excitable HEK293 cells" stably transfected with RyR1 and transiently transfected with the Ca_V_s (and their auxiliary subunits), JPH3 and R-CEPIAer. For both Ca_V_2.1 and Ca_V_2.2, about a third of the cells that produced a cytoplasmic calcium transient, in response to K depolarization, also manifested Ca^2+^ release from the ER (Figure 7E-J), consistent with the idea that Ca_V_2.1/RyR1/JPH3 and Ca_V_2.2/RyR1/JPH3 can both support CICR.

We agree with the referee that it is important to know whether there are functional interactions between JPH3 and JPH4 and had begun such experiments prior to the initial submission of this manuscript. Because JPH3 strongly recruits RyR1 and JPH4 does not, we thought examining the recruitment of RyR1 by mixed junctions would be a good place to start. In agreement with Sahu et al. (2019), we found that JPH3 and JPH4 could intermingle in junctions. These mixed junctions differed dramatically in their ability to recruit RyR1: some recruited nearly as effectively as pure JPH3, whereas others were nearly as ineffective as pure JPH4. We suspect that sorting this out will require a detailed analysis of the JPH3:JPH4 ratio and whether this is uniform, or regionally variable, in individual junctions.

3. The finding that Jph3 and Jph4 selectively slows-down inactivation of Ca_V_2 channels but not Ca_V_1 is novel and highly interesting. For these studies, the authors use Ca^2+^ as charge carrier with high intracellular Ca^2+^ buffering (10 mM EGTA). As Ca^2+^-dependent inactivation (CDI) of Ca_V_2 channels requires global Ca^2+^ elevation, this change in inactivation of Ca_V_2.1 and Ca_V_2.2 likely reflects a change in voltage-dependent inactivation (VDI) as the authors point out. However, CDI of Ca_V_1.2 channels is sensitive to local Ca^2+^ which may be quite strong under these conditions. If JPH were to only affect VDI, then strong CDI could potentially mask underlying differences in VDI of Ca_V_1.2. Have the authors considered using Ba2+ current measurements to further establish the absence of any effect of JPH on VDI of Ca_V_1.2?

We have added results of experiments on the inactivation of Ba^2+^ currents via Ca_V_1.2, Ca_V_2.1 and Ca_V_2.2 in the absence and presence of JPH4 (which had a larger effect on inactivation of Ca_V_2.1 and Ca_V_2.2 than did JPH3). We found that JPH4 somewhat reduced the inactivation of Ba^2+^ currents via Ca_V_1.2. Thus, JPH4 appeared to decrease VDI of Ca_V_1.2 which could have been masked for the Ca^2+^ currents by an increase in CDI, as suggested by the referee. We have not investigated this issue further, but an increase in CDI could result from the presence of endogenous RyRs at the ER-PM junctions (see Point 1 cont'd above). An increased CDI could also be a consequence of the ER-PM junctions per se which produce a restricted space and clustering of channels. However, even for Ba^2+^ currents, JPH4 caused a much larger decrease in the inactivation of Ca_V_2.1 and Ca_V_2.2 than in the inactivation of Ca_V_1.2.

4. Comparing figures 3 and 4, full length JPH3 only modestly slows inactivation of Ca_V_2.1. However, the truncated variant (1-707) exhibits stronger inhibition of inactivation. What causes this difference? Could it be be because JPH3 only partially recruits Ca_V_2.1 to the ER-PM junctions, while the truncated form can act on channels that may be outside of the junctional space? The difference in potency appears to be lesser for Ca_V_2.2, which has a higher colocalization with JPH3 than Ca_V_2.1

We think this is an attractive hypothesis and have added it to the revised manuscript.

5. Statistical analysis comparing colocalization coefficients would be helpful, particularly in inferring rank order of JPH3 preference with different RyR isoforms (line 259). Given the variability in the colocalization coefficients, it is possible that some of the apparent differences may not be statistically significant. For e.g. JPH3:RyR1 versus JPH3:RyR3 or JPH4:RyR2 versus JPH4:RyR1.

The revised manuscript now contains a complete statistical comparison of all the co-localization coefficients, which is included in a section entitled "Raw data and statistics." The statistical significance of the comparisons that the referee asked about are included as part of Figure 5. JPH3:RyR1 versus JPH3:RyR3 is borderline significant (p=0.0248) and JPH4:RyR2 versus JPH4:RyR1 is not significant (p=0.2366).

6. One shortcoming of the present study is that the differential association of JPH/RyR/Ca_V_ channel complexes are probed only in TSA cells under overexpression conditions. While this allows for comparison of the propensity of JPH to recruit different ion channels, it may not reflect the composition of native ER-PM junctions in neurons. This study may be further bolstered by analysis of co-localization of endogenous RyR, JPH, and Ca_V_ channels in neurons.

Defining the composition and function of JPH-containing ER-PM junctions in neurons remains the important goal. However, the results we have described – including the effects of JPH3 and JPH4 on calcium channel function, the identification of Ca_V_2.2 as a novel candidate for supporting CICR in neurons, and the demonstration and partial elucidation of binding interactions between JPH3 and the cytoplasmic domains of RyR1 and RyR3 – represent significant information that we believe merits publication.

Our lab does plan to pursue the analysis of neuronal ER-PM junctions but doing this properly will not be a trivial undertaking. For example, there are already commercially available antibodies for junctophilins, Ca_V_s and RyRs, but commercial antibodies often lack specificity when used for IHC in native tissues. Arguably, antibody specificity can best be tested in subtype-specific knockouts. Alternatively, mice can be generated with knock-ins fusing fluorescent proteins to the target proteins of interest. We hope to establish the collaborations and generate the funding that will allow us to pursue these approaches.